# Measuring Accessibility of Healthcare Facilities for Populations with Multiple Transportation Modes Considering Residential Transportation Mode Choice

**Xinxin Zhou [1,2], Zhaoyuan Yu [1,2], Linwang Yuan [1,2], Lei Wang [1,2] and Changbin Wu [1,2,*]**

1  School of Geography, Nanjing Normal University, Nanjing 210023, China; 181301027@stu.njnu.edu.cn (X.Z.); yuzhaoyuan@njnu.edu.cn (Z.Y.); yuanlinwang@njnu.edu.cn (L.Y.); 181335010@stu.njnu.edu.cn (L.W.)
2  Key Laboratory of Virtual Geographic Environment, Ministry of Education, Nanjing Normal University, Nanjing 210023, China
*  Correspondence: wuchangbin@njnu.edu.cn

**Abstract:** Accessibility research of healthcare facilities is developing towards multiple transportation modes (MTM), which are influenced by residential transportation choices and preferences. Due to differences in travel impact factors such as traffic conditions, origin location, distance to the destination, and economic cost, residents' daily travel presents different residential transportation mode choices (RTMC). The purpose of our study was to measure the spatial accessibility of healthcare facilities based on MTM considering RTMC (MTM-RTMC). We selected the gravity two-step floating catchment area method (G2SFCA) as a fundamental model. Through the single transportation mode (STM), MTM, and MTM-RTMC, three aspects used to illustrate and redesign the G2SFCA, we obtained the MTM-RTMC G2SFCA model that integrates RTMC probabilities and the travel friction coefficient. We selected Nanjing as the experimental area, used route planning data of four modes (including driving, walking, public transportation, and bicycling) from a web mapping platform, and applied the three models to pediatric clinic services to measure accessibility. The results show that the MTM-RTMC mechanism is to make up for the traditional estimation of accessibility, which loses sight of the influence of residential transportation choices. The MTM-RTMC mechanism that provides a more realistic and reliable way can generalize to major accessibility models and offers preferable guidance for policymakers.

**Keywords:** spatial accessibility; multiple transportation modes; residential transportation mode choice; web maps; route planning data; healthcare facility

## 1. Introduction

Spatial accessibility, which is influenced by the type and scale of service facilities, the spatial population redistribution in the demand locations, and travel impedance (such as travel time, distance, or travel expense), plays an essential part in planning urban service facilities and optimizing urban traffic [1–6]. The accessibility of healthcare is a multifaceted field that involves primary healthcare [7], access to healthcare in rural areas [8,9], the cross-border spatial accessibility of healthcare [10], the spatial equity of multilevel healthcare [11], hospital care and emergency medical services [12,13], and mental health in childhood and adolescence [14]. Travel impedance has been widely proven by scholars to be a key factor affecting residents' access to service facilities [15]. Nevertheless, the measurement of travel impedance faces difficulties, so how to obtain more realistic travel impedance in multimodal, dynamic traffic environments?

First, residents' daily travel to healthcare is toward multimodal transportation. Traditional studies have looked towards subjectively choosing specific travel modes [16], but an increasing number of

scholars have begun to improve accessibility models considering the influence of different transport modes [17]. The transportation modes mainly include driving, walking, public transportation, bicycling, and combinations of these; among them, combination modes are usually adopted for long-distance travel [18]. Diverse transportation modes have a significant impact on the traffic route and traffic time [19], such as variations in travel route planning and time consumption in nine European cities and the world's 15 largest metropolises. An accessibility model with multiple transportation modes (MTM) has been proposed to account for realistic variations in travel time and service reliability [20]. Assuming that residents have the same choice probability for each type of transportation mode to different destinations, Mao and Nekorchuk [3] considered driving and public transportation modes to estimate healthcare accessibility. Lin et al. [21] calculated the travel time for primary care providers by car and public transportation based on StreetMap data. Based on the travel impedance data considered in the MTM, many new accessibility models have been developed, such as the multimodal two-step floating catchment area method (2SFCA) [3], a variable-width floating catchment area model [17], multimodal 2SFCA incorporating the spatial access ratio [21], and multimodal accessibility-based equity assessment [22].

Second, the speed of various modes of transportation is not a constant theoretical value. A consistent collection of accessibility studies have calculated travel impedance by simulating a single transportation mode with a theory-based speed based on a geographic information system (GIS) road network dataset, for example, a walking speed of 4 km/h, a biking speed of 10 km/h, and a driving speed on major urban roads of 50 km/h [23–25]. However, the existing literature is heavily based on oversimplified assumptions that transit services operate at deterministic speeds [20]. Travel impedance data can be accessed from open data sources [26,27], including individual trip survey data [3], web mapping services (Google Maps [17], Baidu Maps [28], Amap Maps [29]), and location-based social media data [20], which enable advancements in revealing the characteristics of human activities [30–32]. Georeferenced social media data provide fine-scale and dynamic big data for accessibility research [33]. Web mapping services provide a more accurate approach for obtaining travel impedance data between an origin and destination for MTM [25].

Third, individuals have different preferences for different modes of transportation. There have been fewer experiments on how residential transportation mode choices (RTMC) influence facility accessibility. RTMC refers to the selection of transportation modes by residents based on their activities, travel time restrictions, destinations, and traffic conditions [34–36]. There are multiple transportation modes between an origin–destination pair (OD), also called the demand point $i$ to the supply point $j$. The basic premise of RTMC assumes that people prefer to choose the most convenient and quick transportation mode [20,37], which depends on the "limited rationality" of human decision-making [38]. In real life, traffic facility conditions, traffic location, distance from the destination, and economic cost become influencing factors of RTMC [39]. The theories of RTMC mainly include utility maximization theory [40], satisfaction evaluation theory [41], and effort-accuracy trade-off theory [42]. We need to pay attention to the differences in residents' choices of different traffic modes based on individual factors: the factors that impact traffic for residents, such as the effect of disparities in travel time on diverse transportation modes caused by the degree of urban congestion at different times.

In this paper, we will focus on the RTMC gap and consider the various transportation factors to establish RTMC probabilities and redesign the gravity two-step floating catchment area method (G2SFCA). The RTMC probabilities, which mainly depend on different transportation modes' travel impedance (Figure 1a), illustrate that different transportation modes from an origin location $i$ (demand location) to destination location $j$ (supply location) have different probabilities $w(m_i)$ of being chosen (Figure 1b). We selected the single transportation mode (STM) G2SFCA model as a fundamental model and measured spatial accessibility to facilities with MTM-RTMC. We chose Nanjing as the experimental area based on four types of route planning data for MTM from the web mapping platform; we then applied three G2SFCA models to pediatric medical facilities to measure accessibility.

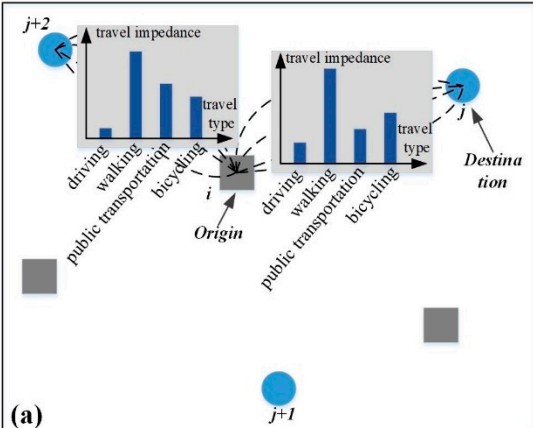 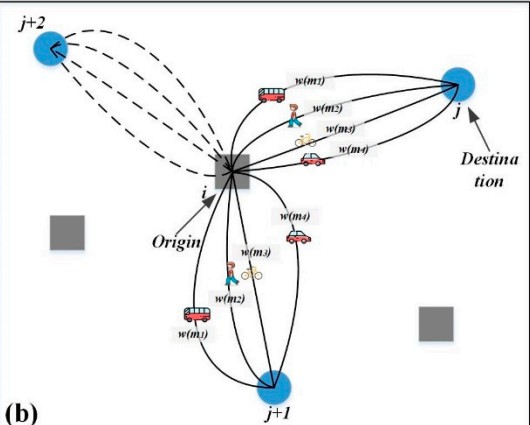

**Figure 1.** Schematic diagram of multiple transportation modes-residential transportation mode choices (MTM-RTMC). The RTMC probabilities, which mainly depend on different transportation modes' travel impedance (Figure 1a), illustrate that different transportation modes from an origin location *i* (demand location) to destination location *j* (supply location) have different probabilities $w(m_i)$ of being chosen (Figure 1b).

The remainder of this paper is organized as follows. Section 2 describes the mathematical accessibility models for the STM G2SFCA model (Section 2.2), the MTM G2SFCA model (Section 2.3), and the MTM-RTMC G2SFCA model (Section 2.4) in detail. Additionally, we introduce two critical parameters in the MTM-RTMC G2SFCA model, including the travel friction coefficient (Section 2.5) and RTMC probabilities (Section 2.6). Section 3 introduces the background of the research area and the primary status of the data used in this research. Section 4 discusses the value of the proposed method and its results for measuring healthcare facility accessibility. In Sections 5 and 6, we generalize the MTM-RTMC mechanism into five types of 2SFCA models and summarize the conclusions of this research.

## 2. Materials and Methods

Several studies have used the 2SFCA method to estimate healthcare accessibility [16,43]. The generalized 2SFCA model is the fundamental accessibility model [44]. With changes in the distance decay functions, the generalized 2SFCA model can evolve into many types, including the enhanced 2SFCA model, the G2SFCA model, the kernel density 2SFCA model, and the Gaussian 2SFCA model [15]. Among accessibility measurement models, the G2SFCA model has been widely used to measure facility spatial accessibility [22]. The G2SFCA model considers the interaction between supply and demand located in different areas [45]. The simulation-based G2SFCA model, which considers the distance decay function as a continuous function, is one of the represented 2SFCA methods [25]. In this section, the generalized 2SFCA model, which is the basis of Sections 2 and 4.1, is reviewed first, and three G2SFCA models are introduced to illustrate the implementation process for the improvement and design of the MTM-RTMC G2SFCA model.

### 2.1. Review of the Traditional Generalized 2SFCA Model

In the accessibility methodology field, we can categorize the accessibility models into two categories, place-based (i.e., residential community) and individual-based (i.e., residential individual) accessibility models classified by the number of residents studied [15]. The place-based accessibility measures mainly include cumulative opportunity models and gravity models [46] and the 2SFCA [47], and evaluate the opportunities from demand locations to surrounding facilities considering the travel impedance [48]. Radke and Mu [49] presented earlier versions of the floating catchment area approach, and the 2SFCA was further improved by Luo and Wang [47]. Wang [16] summarized the general form of the 2SFCA model, the generalized 2SFCA model. First, the generalized 2SFCA model defines the catchment

of the supply location $j$ as an area comprised of all demand locations $k$ within a threshold travel impedance $d_0$, and calculates the supply-demand-ratio $X_j$ within the catchment area (Equation (1)). Second, the generalized 2SFCA model searches all $j$ within $d_0$ from $i$ and summarizes the ratios $X_j$ (Equation (1)) [16]. The population scale factor $V_j$ is the total serviceable population after the combined effect of the distance decay functions $f(d_{kj})$ (Equation (2)). $V_j$ can illustrate differences in the serviceable population scale from different supply points [50]. The global accessibility value of all demand points can be calculated from Equation (3).

$$A_i(Generalized) = \sum_{j=1}^{M} X_j = \sum_{j=1}^{M} \frac{E_j * f(d_{ij})}{V_j} \tag{1}$$

$$V_j = \sum_{k=1}^{N} P_k * f(d_{kj}) \tag{2}$$

$$A(Generalized) = \sum_{k=1}^{N} A_i(Generalized) \tag{3}$$

where
　　$A_i(Generalized)$—the spatial accessibility of the demand point $i$ calculated by the generalized 2SFCA model;
　　$A(Generalized)$—all demand points sum value of spatial accessibility calculated by the generalized 2SFCA model;
　　$E_j$—the service resource supply capacity of supply point $j$;
　　$d_{ij}$—the travel impedance from demand point $i$ to $j$;
　　$M$—the total number of supply points;
　　$N$—the number of demand points;
　　$P_k$—population number at demand point $k$;
　　$V_j$—the population-scale factor of $j$;
　　$X_j$—the supply-demand ratio of $j$.
　　The distance decay functions $f(d_{ij})$ and $f(d_{kj})$ of the generalized 2SFCA model can be presented as Equation (4); $g(d_{ij})$ is usually a specific constant.

$$f(d_{ij}) = \begin{cases} g(d_{ij}), & d_{ij} \le d_0 \\ 0, & d_{ij} > d_0 \end{cases} \tag{4}$$

where
　　$d_0$—the travel impedance effective threshold between $i$ and $j$.

### 2.2. Review of the Traditional STM G2SFCA Model

　　The gravity model and the 2SFCA model belong to the same theoretical framework [47]. Both of them consider the influence of destination scale, standpoint scale, and distance relationship between destination and standpoint on accessibility. The STM G2SFCA model, also known as the potential model or the potential energy model, is derived from Newton's law of universal gravitation and was proposed by Hansen [51]. Hansen introduced the concept of spatial accessibility when analyzing the population distribution of the urban population and the spatial accessibility indicators of residential land. Hansen also offered a model to calculate spatial accessibility, which was evaluated using the potential indicators [52].

The basic form of the STM G2SFCA model ($A_i(stm)$, $A(stm)$) is consistent with the generalized 2SFCA model except for the distance decay function $f(d_{ij})$. The $f(d_{ij})$ of the STM G2SFCA model can be generalized as Equation (5).

$$f(d_{ij}) = \begin{cases} d_{ij}^{-\beta}, & d_{ij} \leq d_0 \\ 0, & d_{ij} > d_0 \end{cases} \tag{5}$$

where

$\beta$—the coefficient of travel friction: the distance-decay parameter.

### 2.3. Designing the MTM G2SFCA Model

MTM provides a more realistic accessibility representation than single-mode methods [21]. Langford, Higgs, and Fry [43] proposed a multimodal 2SFCA method by incorporating both public and private transportation modes using dedicated network datasets. In this paper, we adopt the MTM mechanism to rebuild the population-scale factor $V_j$ (Equation (6)) and distance decay function $f(d_{kj}(m_r))$ (Equation (8)) first, and acquire the accessibility of MTM (Equation (7)) second. In Equation (8), the travel friction coefficients $\{\beta(m_1), \beta(m_2), \ldots \ldots \beta(m_r)\}$ vary by transportation mode, which can specify the different travel impedance decay effects of accessibility [3]. More details of the travel friction coefficients $\{\beta(m_1), \beta(m_2), \ldots \ldots \beta(m_r)\}$ are provided in Section 2.5. Because $R$ types of transportation modes are considered in Equations (6) and (7), the MTM results need to be divided by $R$ to avoid double counting. Summing all $A_i(mtm)$, we can obtain $A(mtm)$, the global value of the research area (Equation (9)).

$$V_j = \frac{1}{R} * \sum_{k=1}^{N} \sum_{r=1}^{R} P_k * f(d_{kj}(m_r)) \tag{6}$$

$$A_i(mtm) = \frac{1}{R} * \sum_{j=1}^{M} \sum_{r=1}^{R} \frac{E_j * f(d_{ij}(m_r))}{V_j} \tag{7}$$

$$f(d_{ij}(m_r)) = \begin{cases} (d_{ij}(m_r))^{-\beta(m_r)}, & d_{ij}(m_r) \leq d_0(m_r) \\ 0, & d_{ij}(m_r) > d_0(m_r) \end{cases} \tag{8}$$

$$A(mtm) = \sum_{k=1}^{N} A_i(mtm) \tag{9}$$

where

$A_i(mtm)$—the MTM G2SFCA model's spatial accessibility of $i$;

$A(mtm)$—all demand points sum of MTM G2SFCA model's spatial accessibility value;

$m_r$—the $r$ type transportation mode, $r \in (0, R]$. Transportation modes include driving, walking, public transportation, and bicycling;

$\beta(m_r)$—the travel friction coefficient of transportation mode $m_r$;

$d_{ij}(m_r)$—the travel impedance for transportation mode $m_r$ from $i$ to $j$;

$d_0(m_r)$—the travel impedance effective threshold of transportation mode $m_r$ between $i$ and $j$.

### 2.4. Designing the MTM-RTMC G2SFCA Model

There is an underlying assumption that residents have the same probabilities of different situations in $A_i(mtm)$, which deviates from the reality of transportation. Therefore, we employed the RTMC probabilities $w_k(m_r)$ for transportation model $m_r$ to get closer to the reality of transportation (see Equations (10) and (11)). Among them, the $w_{ij}(m_r)$ that satisfies the constraints is $\sum_{r=1}^{R} w_{ij}(m_r) = 1$.

$$V_j = \sum_{k=1}^{N} \sum_{r=1}^{R} w_{kj}(m_r) * P_k * f(d_{kj}(m_r)) \tag{10}$$

$$A_i(mtm - rtmc) = \sum_{j=1}^{M} \sum_{r=1}^{R} \frac{w_{ij}(m_r) * E_j * f(d_{ij}(m_r))}{V_j}$$
$$st. \sum_{r=1}^{R} w_{ij}(m_r) = 1 \tag{11}$$

To understand the master equation of the MTM-RTMC G2SFCA model (Equation (11)) more intuitively, the expansion form of $A_i(mtm - rtmc)$ is shown in Equation (12) without considering the influence of $d_0(m_r)$. Summing all $A_i(mtm - rtmc)$, we can obtain $A(mtm - rtmc)$, the global value of the research area (Equation (13)). In the MTM-RTMC G2SFCA model, the process for calculating the critical parameter $w_{ij}(m_r)$ is shown in Section 2.6.

$$A_i(mtm - rtmc) = \sum_{j=1}^{M} \left( \frac{w_{ij}(m_1) * E_j}{d_{ij}(m_1)^{\beta(m_1)} * V_j} + \frac{w_{ij}(m_r) * E_j}{d_{ij}(m_r)^{\beta(m_r)} * V_j} + \ldots + \frac{w_{ij}(m_R) * E_j}{d_{ij}(m_R)^{\beta(m_R)} * V_j} \right)$$
$$st. \sum_{r=1}^{R} w_{ij}(m_r) = 1 \tag{12}$$

$$A(mtm - rtmc) = \sum_{i=1}^{N} A_i(mtm - rtmc) \tag{13}$$

where

$A_i(mtm - rtmc)$—the MTM-RTMC G2SFCA model's spatial accessibility of $i$;

$A(mtm - rtmc)$—all demand points sum of the MTM-RTMC G2SFCA model's spatial accessibility value;

$w_{ij}(m_r)$—the RTMC probabilities of transportation mode $m_r$ from $i$ to $j$.

### 2.5. Travel Friction Coefficient β

The travel friction coefficient β requires additional data and work to define and might be region-specific. Predefining the travel friction coefficient β for different transportation modes is also crucial in the research of predecessors (Table 1) because it determines the rate of $d_{ij}$ decay to calculate MTM accessibility and MTM-RTMC accessibility [53]. A larger β value suggests that residents are more discouraged by long-travel times when seeking healthcare facilities and thus have a higher tendency to settle for facilities in nearby locations [47]. When studying bus and car modes, Mao and Nekorchuk [3] ignored the influence of travel friction. They observed that ignoring the impact of residents' choices of transportation modes would lead to deviation from more realistic results. The travel friction coefficient β reveals the general effect of travel impedance decay on spatial interactions. A greater β indicates a faster distance decay effect such that movements are more likely to be impeded due to spatial segregation [54].

Regardless of the impact of individual factors on the choice of individual residents' transportation modes, the overall travel of the residents' group follows the fundamental assumptions as follows. (1) Residents can choose one of the transportation modes. (2) The basic principles of RTMC are inclined toward a transportation mode with a short-travel time and low economic cost. (3) The general average speed characteristics of the four main transportation modes are walking at 4 km/h (lowest speed), bicycling at 8 km/h (lower speed), public transportation at 40 km/h (medium speed), and driving at 70 km/h (high speed) [23–25]. (3) Considering the travel friction coefficient β in the distance decay functions that enhance and suppress the travel impedance, β = 1 is the critical value point, with the travel impedance threshold being 1 kilometer as the boundary dividing travel into the short and long distances (Figure 2) (for travel impedance, the distance is selected as $d_{ij}$; refer to the content: choose kilometer as the dimension, and the basis is discussed in detail in Section 4.1). (4) In short-distance travel, the travel friction coefficient β of the distance decay functions has an enlarging effect on travel impedance. In contrast, in long-distance travel, the travel friction coefficient β of the distance decay functions has a shrinking effect on the travel impedance (Figure 2). When the distance to a facility is minimal, that facility has an absolute advantage over other more remote service facilities.

Adjacent service facilities have a leading role in neighboring demand points and are therefore worthy of attention.

**Table 1.** Different travel friction coefficients $\beta$ in the research of predecessors.

| Researchers | Selected $\beta$ Value | Transportation Mode | Travel Impedance | Research Area | Scientific Question |
|---|---|---|---|---|---|
| [55,56] | 2.0 | Driving | Travel time | Rudong County, Jiangsu Province, China | The accessibility of health care facilities |
| Wang and Tang [44] | 0.6 to 1.8 | Unqualified, simulation-based on GIS for obtaining the shortest travel distance | Travel distance | Chicago, America | The highest equality of accessibility |
| Yao et al. [57] | 1.0 | Unqualified | Travel time | four districts (Chibuto, Chokwè, Guíjà, and Mandlakaze) of Gaza Province in southern Mozambique | Utilization of sexual and reproductive health (SRH) services |
| Tao, Cheng, Dai, and Rosenberg [45] | 0.6 to 1.4 | Unqualified, simulation-based on GIS for obtaining the shortest travel time | Travel time | Beijing, China | Spatial optimization of residential care facility locations |
| Barona and Blaschke [58] | 1 | Unqualified, travel distance | Travel distance | Quito, Ecuador | Healthcare accessibility and socioeconomic deprivation |
| Zhang, Cao, Liu, and Huang [53] | 0.8 | Unqualified, simulation-based on GIS for obtaining the shortest travel distance | Travel distance | Hongkong SAR, China | A multi-objective optimization approach for healthcare facility location-allocation problems |
| Zhu, Huang, Shi, Wu, and Liu [54] | 1.0 | Unqualified, travel distance | Travel distance | China | Inferring spatial interaction patterns |
| Hu and Downs [59] | 0.602 | Unqualified, based on Google Maps Distance Matrix API | Travel time | Tampa Bay Region, Florida, America | Space-time job accessibility |
| Chen and Jia [15] | 1.5 and 2.0 | Unqualified, the shortest path O-D cost matrix between demand points and supply points using the Network Analysis module in ArcGIS 10.4. | Travel distance | Arkansas, America | Supplemental Nutrition Assistance Program (SNAP) authorized food retailers in the state |

We can conclude the following principles:

(1) In short-distance travel, the travel times of the four main transportation modes are similar. The residents tend to adopt low-cost transportation modes, such as walking or cycling, so the travel speed and the travel friction coefficient $\beta$ have a negative correlation. In short-distance travel, the travel friction coefficient β in the high-speed transportation mode offers a relatively smaller enhancement of the travel impedance as the speed of the transportation mode increases. So, the travel friction coefficient β in the high-speed transportation mode is relatively low.

(2) For long-distance travel, the travel time of the four main transportation modes varies greatly. The diversity of alternative transportation modes gradually decreases, and residents tend to choose faster transportation modes, which can ensure shorter transit times. As the speed of the transportation mode increases, the travel friction coefficient $\beta$ of the high-speed transportation mode has a relatively smaller effect on suppressing the travel impedance, so the travel friction coefficient $\beta$ of the high-speed transportation mode is relatively low.

In the MTM G2SFCA model and MTM-RTMC G2SFCA model, we assigned the travel friction coefficients as 1.4, 1.2, 1.0, and 0.8 for walking, bicycling, public transportation, and driving, respectively, which conforms to the suppressing effect on the speed of transportation modes.

### 2.6. Residential Transportation Mode Choice Probabilities $w_k(m_r)$

Many of the factors that affect the RTMC can be divided into macro and micro aspects (Figure 3). The macro factors include urban area characteristics, municipal economic level, and development status of urban transportation facilities, while the micro factors include individual resident attributes, family attributes, accessibility, punctuality, comfort, safety, travel experience, real-time traffic information,

travel purpose, departure time urgency, travel distance, etc. Differences in the macro factors lead to overall differences between urban regions. The differences in micro factors lead to a diversity of transportation modes for regional residents; for example, because the travel budget of high-income residents is higher, the ownership rate of private cars and the ratio of travel by driving will be higher. The RTMC presents a combination probability distribution, and the overall trend of this distribution is consistent with the whole urban travel cost context.

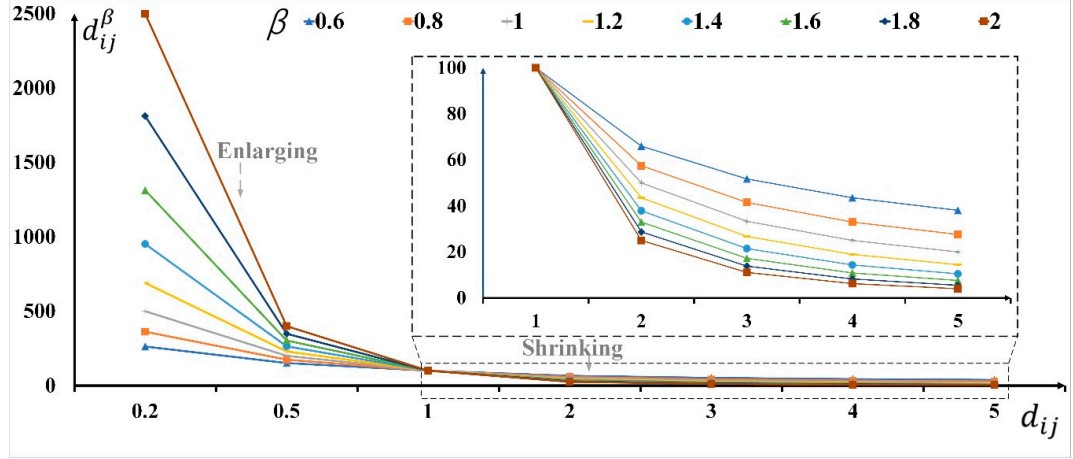

**Figure 2.** Schematic effect of different travel friction coefficients $\beta$ in the line chart.

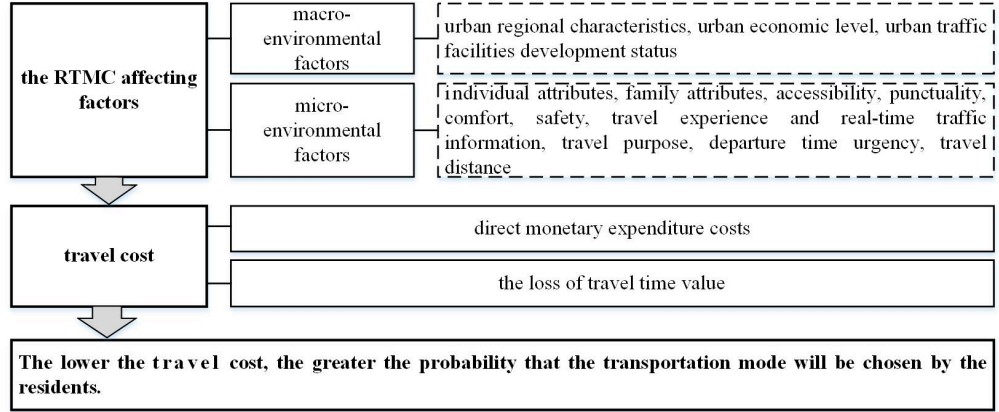

**Figure 3.** The RTMC theoretical model diagram.

The travel cost refers to the total currency performance that residents invest in the travel process and is reflected in monetary and non-monetary expenditures. The travel cost applied to residents selecting a specific transportation mode can sum into the direct financial expenditure and the loss of travel time during the entire travel process. The loss of travel time value is defined as the currency performance of the individual's time consumed during the entire travel process. The principle of a shadow price is that if the time consumed had instead been invested in production activities, a commodity of particular value would be created; in other words, the amount of value lost refers to the amount of money lost from investing a certain amount of time in non-productive activities [60]. Residents tend to choose a transportation mode with lower travel costs (Figure 3).

The travel cost is mainly affected by traffic time, traffic distance, and the economic coefficient or urgency. The larger the traffic time and traffic distance are, the higher the travel cost [40]. We calculated RTMC probabilities $w_k(m_r)$ considering the affecting factors. The RTMC probability calculation explicitly includes four steps (Figure 4). The input data are the route planning data of MTM, and the

output result is $w_k(m_r)$ of different OD. We relied on the Z-score standardization and Softmax function [61] dimension reduction to calculate the RTMC probabilities.

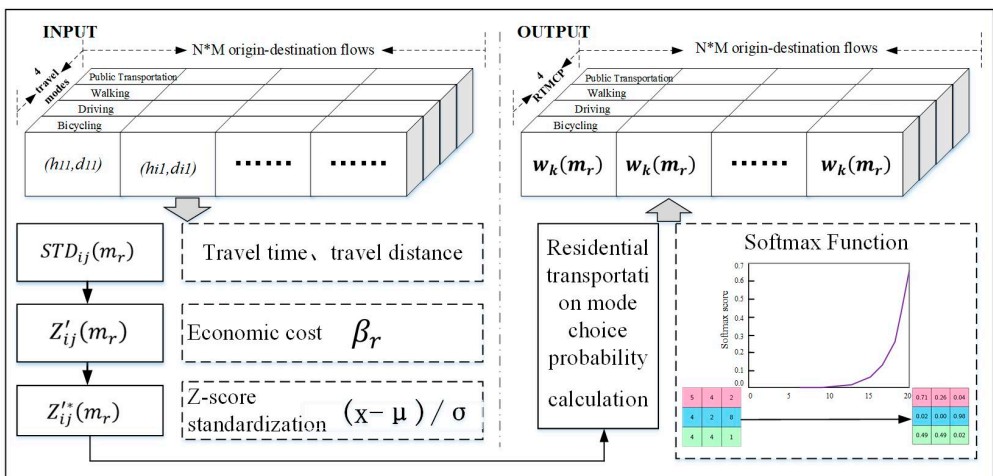

**Figure 4.** The RTMC probability calculation schematic diagram.

Step 1: Building $STD_{ij}(m_r)$ (Equation (14)). The $STD_{ij}(m_r)$ is characterized by the degree of physical connection in geospatial space.

$$STD_{ij}(m_r) = \sqrt{h_{ij}(m_r) * d_{ij}(m_r)} \tag{14}$$

where

$h_{ij}(m_r)$—the travel time for transportation mode $r$ from $i$ to $j$;

$d_{ij}(m_r)$—the travel distance for transportation mode $r$ from $i$ to $j$.

Step 2: Building $Z'_{ij}(m_r)$ (Equation (15)). $\delta_r$, which has a unit of dollar/(person.km), indicates the economic cost factor for integrating different transportation modes. The economic cost factor refers to the trade-offs between uses of resources [39,62]. The design of $\delta_r$ has a major adjustment impact on the calculation of the RTMC probabilities. We used fixed vehicle costs and variable vehicle costs as users' money costs. Therefore, the indicators are average car: $0.15, diesel bus: $0.08, bike: $0.03, walk: $0.01 [39,62]. For ease of calculation, we increased $\delta_r$ with 1 and established $\delta_r$ as {*Public Transportation = 1.08, Walking = 1.01, Driving = 1.15, Bicycling = 1.03*}. $\delta_r$ is a negative indicator, and the larger it is, the lower the RTMC probability value.

$$Z'_{ij}(m_r) = \frac{1}{STD_{ij}(m_r) * \delta_r} \tag{15}$$

Step 3: Z-score standardization (Equation (16)). The Z-score standardization method, which reduces the deviation of the selection probability difference caused by the large difference in OD, was selected for standardization. The Z-score standardization method converts multiple sets of data into unitless $Z'^*_{ij}(m_r)$. The scores make the data standards uniform, improve data comparability, and weaken data interpretability.

$$Z'^*_{ij}(m_r) = \frac{Z'_{ij}(m_r) - \mu}{\sigma} \tag{16}$$

where

$\sigma$—the standard deviation of the overall data;

$\mu$—the mean of the overall data.

Step 4: RTMC probability $w_{ij}(m_r)$ in Equation (17). The Softmax function, also known as the normalized exponential function, is a generalization of logic functions in probability theory and related

fields. It can contain any *K* dimension of vector *Z* and "compress" it to another *K* dimension vector $\sigma(Z)$. Therefore, every element has a range in (0, 1), and the sum of all elements is equal to 1. The Softmax function is a gradient log normalization of a finite item discrete probability distribution, which has a wide range of applications in a variety of probability-based multiclassification problem methods.

$$w_{ij}(m_r) = P(Z_{ij}'^*(m_r)) = \frac{e^{Z_{ij}'^*(m_r)}}{\sum_{r=1}^{R} e^{Z_{ij}'^*(m_r)}}$$
$$\text{st. } \sum_{r=1}^{R} w_{ij}(m_r) = 1 \tag{17}$$

among them, $r \in (0, R]$.

For prominent expression of the calculation process for RTMC probabilities, we selected two empirical OD flows, Yujinli Community to Nanjing Children's Hospital of Guangzhou Office (NCHGZ) (Figure 5a) and Yujinli Community to Yifu Hospital Affiliated to Nanjing Medical University (YHANMU) (Figure 5b), to request the recommended paths of MTM from Amap Maps (point-in-time: 8 October 2019). The detailed travel distances and time results can be seen in Table 2. We calculated RTMC probabilities $w_k(m_r)$ of the selected two empirical OD flows, and the $w_k(m_r)$ results are {*Public Transportation = 0.052, Driving = 0.285, Walking = 0.069, Bicycling = 0.593*} and {Public Transportation = 0.193, Driving = 0.639, Walking = 0.039, Bicycling = 0.13}. Field visits show that Yujinli's location is far from the boarding point of public transportation, so Yujinli's short-distance trips are mainly via bicycle, while Yujinli's long-distance trips are via automobile, which conforms to the actual RTMC.

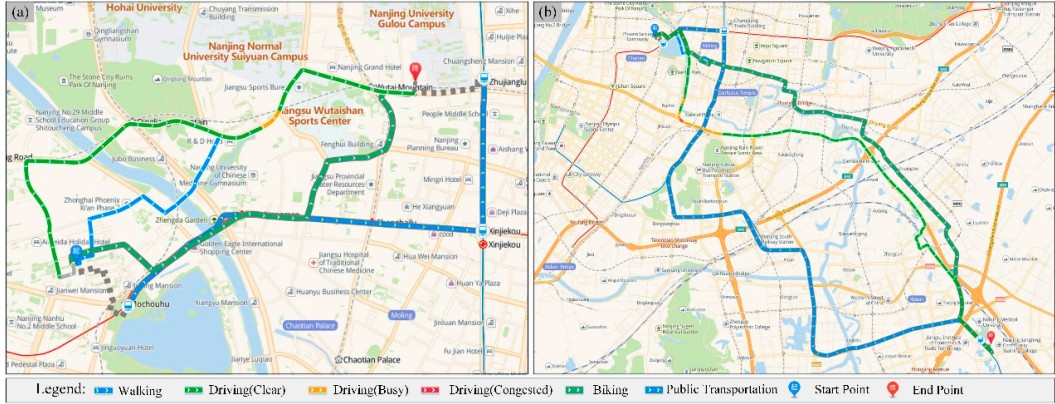

**Figure 5.** The MTM trajectories of two selected empirical origin–destination pair (OD) flows. Figure 5a is short trip from Yujinli Community to NCHGZ, and Figure 5b is long distance trip from Yujinli Community to YHANMU.

**Table 2.** The $d_{ij}$ and $f(d_{ij})$ logbooks for two empirical OD flows (from Yujinli to NCHGZ and to Yifu Hospital Affiliated to Nanjing Medical University (YHANMU)).

| $d_{ij}$ | Unit | Yujinli to NCHGZ | | | | Yujinli to YHANMU | | | | Coefficient of Variation |
|---|---|---|---|---|---|---|---|---|---|---|
| | | W | B | PT | D | W | B | PT | D | |
| travel time | minute | 42 | 15 | 31 | 14 | 321 | 120 | 71 | 75 | 1.1769 |
| | hour | 0.71 | 0.25 | 0.52 | 0.23 | 5.35 | 2.0 | 1.18 | 0.75 | 1.2418 |
| travel distance | meter | 3200 | 3400 | 4500 | 3900 | 21,100 | 23,000 | 28,600 | 23,100 | 0.7949 |
| | kilometer | 3.2 | 3.4 | 4.5 | 3.9 | 21.1 | 23 | 28.6 | 23.1 | 0.7949 |
| | $\beta$ | 1.4 | 1.2 | 1.0 | 0.8 | 1.4 | 1.2 | 1.0 | 0.8 | |
| $f(d_{ij})$ | minute | 0.0053 | 0.0388 | 0.0323 | 0.1211 | 0.0099 | 0.0217 | 0.0330 | 0.0316 | 0.9827 |
| | hour | 1.6152 | 5.2780 | 1.9231 | 3.2405 | 0.2614 | 0.5743 | 0.8760 | 1.2588 | 0.8814 |
| | meter | 0.0000 | 0.0001 | 0.0002 | 0.0013 | 0.0003 | 0.0003 | 0.0003 | 0.0003 | 1.1437 |
| | kilometer | 0.1962 | 0.2303 | 0.2222 | 0.3366 | 0.0872 | 0.0814 | 0.0684 | 0.0811 | 0.6020 |

*Comments*: W: *walking*, B: *bicycling*, PT: *public transportation*, D: *driving*.

### 3. Study Area and Data

*3.1. Study Area*

With an area of 6587 km$^2$ and a population of 8,300,000, Nanjing (118°18' N, 31°14' E, 119°18' N, 32°36' E) is one of the most significant cities in China (Figure 6). Nanjing, the capital of Jiangsu Province and the essential national gateway city for the central and western regions of the Yangtze River Delta, is a famous historical and cultural city. In terms of population, as of 2016, the urbanization rate of Nanjing was 82.29%. In terms of transportation, various transportation forms coexist and are numerous. The number of buses and trolleybuses was 8395, and the number of taxis was 14,239. The total length of the Nanjing metro was 381 km, ranking fifth in length worldwide at the end of 2015. Additionally, the number of personal vehicles in Nanjing reached 2.54 million, and the number of shared bicycles and e-bikes was as high as 650,000 and 3 million, respectively, at the end of 2018 (http://tjj.nanjing.gov.cn/tjxx/201904/t20190402_1495115.html). Finally, the traffic road network of Nanjing is at the forefront for China, with a per capita road area of 21.81 square meters, far exceeding the national average of 15.6 square meters [63]. In terms of medical services, the Nanjing medical and healthcare system is ideal, comprehensive medical resources are abundant, and medical and health conditions rank second only to Shanghai and Beijing. There are 241 public hospitals in Nanjing, of which 22 are level-three hospitals. Although the average number of pediatricians per 1000 people in Nanjing, being 0.67, is higher than the average for China, it is still far below the number for the principal developed countries, whose ratio reaches a standard of 0.85–1.3, which shows a significant gap of pediatric resources. Therefore, considering the population, transportation, and medical services factors, Nanjing was selected as a metropolitan research area with strong typicality. The results thus have reference value for the medical service resource planning of China's first-tier and second-tier metropolises.

To facilitate the research and reduce the influence of the modifiable areal unit problem [64], we used the street block data of Nanjing as a spatial statistical unit. The street block data are the spatial framework data for realizing unified cadastral management, and their division was comprehensively considered based on organizational factors, natural geographical conditions, and public facilities [65]. There are 2265 street blocks in Nanjing. The street blocks in central urban areas demonstrate small and dense areas, with some being smaller than 1 km$^2$. In contrast, the street blocks in peripheral urban areas are large areas and sparsely populated, with some being more than 20 km$^2$. Therefore, using street block data as a research unit can better characterize urban morphological characteristics. It can achieve the essential morphological characteristics of "small difference within the class and large difference between classes."

*3.2. Data*

3.2.1. Route Planning Data of Multiple Transportation Modes

The route planning API of web maps is a new type of travel cost calculation [25]. Web maps provide open-access route planning data for several independent or mixed transportation modes, including driving, walking, and cycling [66]. This paper selected Amap Maps, which is one of the most popular web mapping platforms in mainland China, as a data source. The Amap Maps route planning service offers real-time navigation by producing tailor-made travel plans for users based on destination, departure, and path policy settings that combine real-time traffic data with helping users bypass congested sections and enjoy a more intimate and user-friendly travel experience [67]. The Application Programming Interface (API) is convenient, only requiring the latitude/longitude coordinates of the origin points and destination points. The API-returned results (for requesting code, see supplementary materials) allowed us to directly acquire detailed route files, including specific routes, segmented routes, and corresponding travel costs (e.g., time and distance). The API-returned results are the whole-path time and the distance cost. The travel time and distance by web maps API is a historical average, which gives useful and credible predictions for research purposes. These returned

values are more accurate because they consider the traffic conditions and congestion time loss using real location data [26]. For ease of expression, we selected an empirical provider location, NCHGZ, to show the travel time and distance from different street blocks requested from the route planning API of Amap Maps (point-in-time: 8 October 2019) (Figure 7). Figure 7(a1,a2,a3,a4) present that the travel distance of different transportation modes to NCHGZ is different, which results from that different transportation modes choose different kinds of travel routes. The travel time variability of different transportation modes has apparent differences, shown in Figure 7(b1,b2,b3,b4). The most variability or fastest way is in driving, followed by public transportation, while walking and bicycling have evolved as a series of concentric rings. The above illustrates that different transportation modes have a significant impact on travel time and distance.

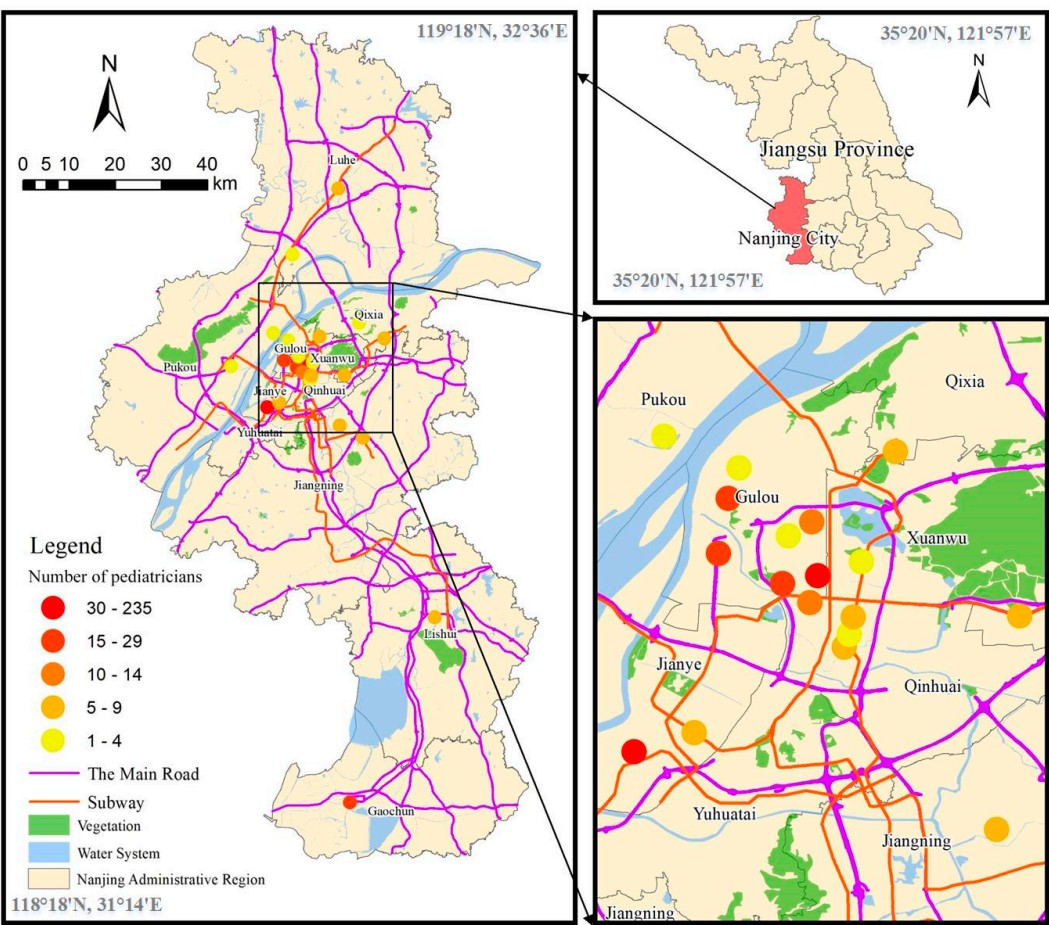

**Figure 6.** The study area.

### 3.2.2. Child Population Spatial Distribution Data

Fine-scale population spatial distribution data are one of the essential indicators for realizing spatiotemporal accessibility [29]. In past studies, the demographic data of each administrative unit have been commonly used to represent the population directly. The shortcomings were large data granularity, discrete spatial distribution, and low accuracy. The development of information and communication technology has provided technical support for obtaining more accurate spatial population distribution data [68]. We adopted the statistical thermal map of the Suitable for Travel Platform to obtain spatial population distribution data. The average value of WeChat thermal map data within this period was used to indicate the regional population distribution.

To accurately obtain the spatial distribution data for the population of children in Nanjing from the Suitable for Travel Platform, we should inspect the definition of a child and adopt the proportional

conversion method. The age division defines the concept of a child as under the age of fourteen. At the end of 2017, the census population from 0 to 14 years old was 904,000 in Nanjing, and the proportion of children in the population was 10.85%. The population data requested from the Suitable for Travel Platform in Nanjing returned a population of 7,769,000, which is slightly lower than the number for the resident population from statistical sources, and thus this is considered to be a reasonable range. Therefore, the number of children in Nanjing was obtained by multiplying 7,769,000 by 10.85%, and the resulting number of children for use in our study was 843,200. To facilitate spatial display, we mapped the child population spatial distribution data on fishnet cells using the natural breaks classification method [69] (Figure 8). The spatial child population (for details, see supplementary materials) is concentrated in the main residential areas of Nanjing, with high-value point-like and uneven spatial distribution. The hotspots are centralized in the regions along with the arterial networks of the main urban area.

### 3.2.3. Pediatric Clinic Services Data

Due to the particular features of pediatrics, pediatric clinic service (PCS) resources are facing high scarcity, especially in developing countries [70,71]. In 2016, the State Health and Family Planning Commission of China issued guidelines for strengthening the reform and development of children's medical and healthcare services (http://www.mohrss.gov.cn/SYrlzyhshbzb/shehuibaozhang/zcwj/yiliao/201606/t20160601_241098.html). The spatial accessibility of PCS is one of the typical and meaningful issues within research on the accessibility of healthcare services [72–74], which prompted us to choose PCS as the research object.

Twenty-six hospitals in Nanjing have established pediatric services (Appendix A Table A1), and the pediatric services of different hospitals currently have significant differences in the level of services they provide. To effectively measure the pediatric scale of various hospitals and consider the availability and accuracy of data, we estimated the pediatric scale of hospitals by the number of pediatricians. The number of pediatricians was obtained from the Good Doctor website (https://haoping.haodf.com/keshi/3030000/faculty_jiangsu_nanjing.htm). The statistical results are expressed in the form of spatialized drawings (Figure 6). The results showed that the total number of pediatric doctors in Nanjing is 603. Overall, compared with 904,000 children, the average number of pediatricians per 1000 people is approximately 0.67. According to data released by the 2015 China Health Statistics Yearbook, in the past five years, there has been an average of 0.43 pediatricians per 1000 children. Although the average number of pediatricians per 1000 people in Nanjing is higher than the average for China, it is still far below the number for the principal developed countries, whose ratio reached a standard of 0.85–1.3.

As the spatial distribution shows, hospitals containing pediatrics are mainly concentrated in the Gulou District, which is the core urban area of Nanjing and has many vital departments, educational resources, and commercial centers.

### 3.3. Ethical Consideration

Our research data, regarding georeferenced social media data, include route planning data of MTM, child population spatial distribution data, and pediatric clinic services data. The crawler program from the Amap Maps API (https://lbs.amap.com/) crawled the route planning data of the MTM. The child population spatial distribution data were obtained from the Suitable for Travel Platform (http://c.easygo.qq.com/eg_toc/map.html). The pediatric clinic services data were manually collected from the Good Doctor website (https://haoping.haodf.com/). The increasing use of georeferenced social media data has brought further privacy challenges, as well as some other social, legal, and ethical issues [75]. Here, we need to declare that all data which were collected from the internet are open source, ethically free, and privacy-free. In addition, the openness of data acquisition brings advantages for our research method to be extended to other cities.

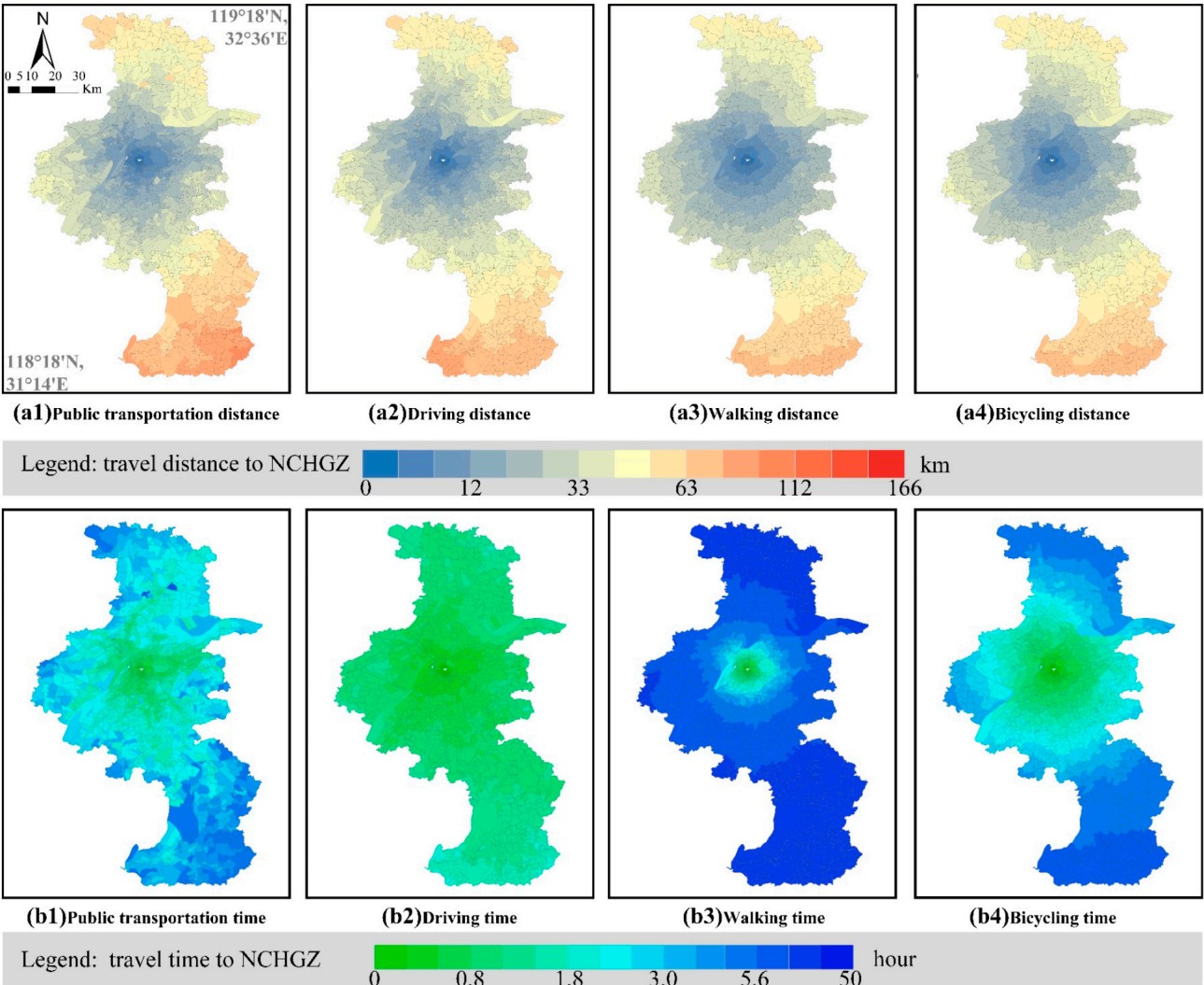

**Figure 7.** MTM time and distance from different street blocks to Nanjing Children's Hospital of Guangzhou Office (NCHGZ) (high-resolution figure can be seen from https://figshare.com/s/7fc7a00e868a9c71ac37).

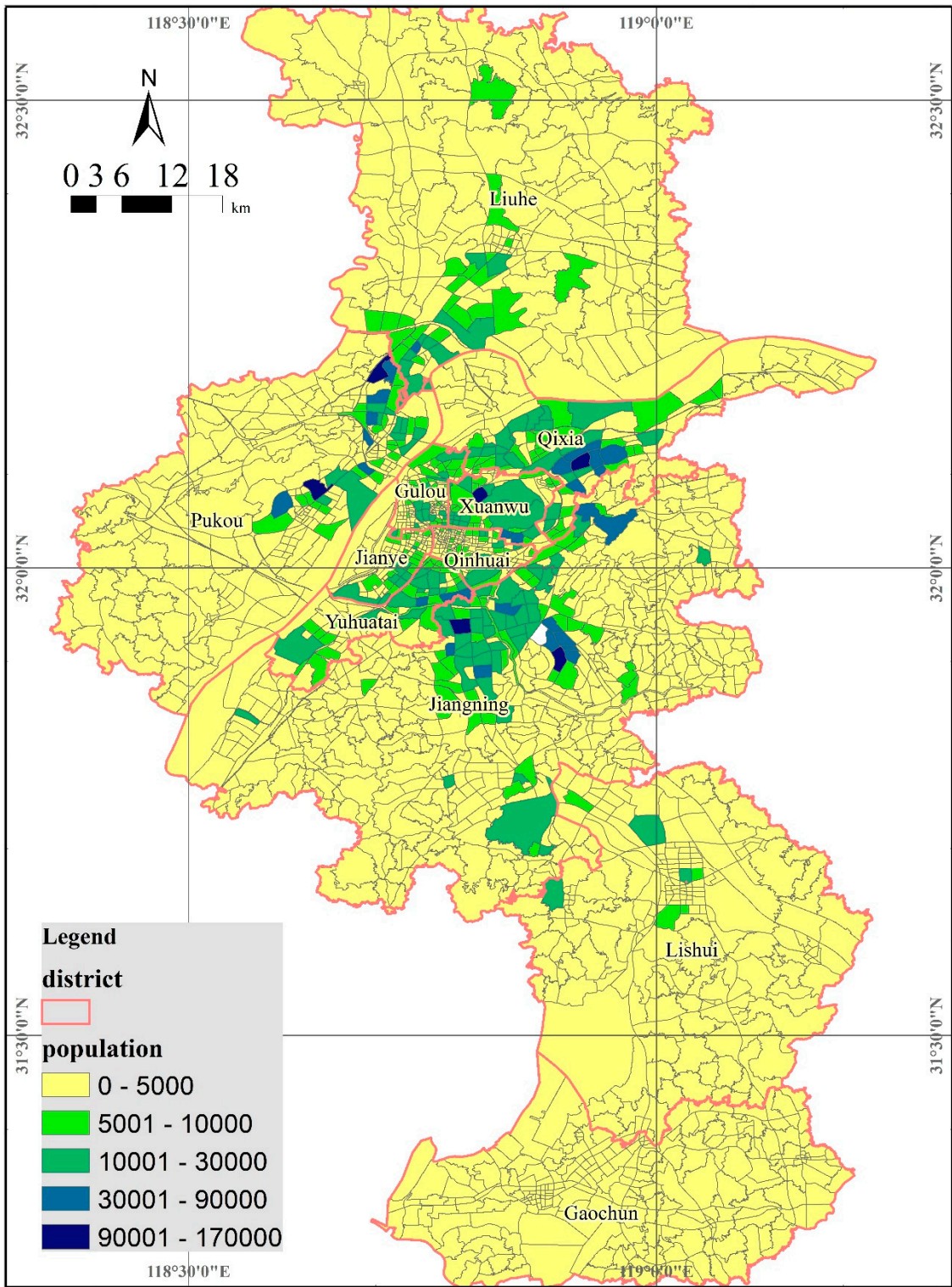

**Figure 8.** The spatial distribution map of Nanjing's child population (high-resolution figure can be seen from https://figshare.com/s/7fc7a00e868a9c71ac37).

## 4. Results

The experimental process was as follows. We designed a crawler program to obtain data and calculate the spatial accessibility of pediatrics based on the three G2SFCA models (STM, MTM, and MTM-RTMC). The data storage software selected in this experiment was MongoDB 2.7.0, the model development language was Python 3.5, and the GIS mapping software was ArcGIS 10.2. In the process of

model calculation, we need to clarify what $d_{ij}$ refers to and its dimension (see Section 4.1). We compared the differences between the $V_j$ of the three models, and the accessibility results of the three models were compared to illustrate the impact of model improvement on accessibility.

### 4.1. $d_{ij}$ Establishment

The content and dimension of $d_{ij}$ in distance decay functions $f(d_{ij})$ have a significant impact on the calculation of accessibility. Scholars have diverse and particular subjectivity about the referred content and dimension of $d_{ij}$ (Table 1). In most 2SFCA models, the travel impedance effective threshold $d_0$ and the distance impedance coefficient $\beta$ are mathematically dependent, and so when $d_0$ is determined, $\beta$ should be adjusted accordingly (Chen and Jia, 2019). To discuss the influence of $d_{ij}$, we again selected the two empirical OD flows: Yujinli to NCHGZ (Figure 5a, relatively close), and Yujinli to YHANMU (Figure 5b, relatively distant). The analysis results (see Table 2) show that the coefficient of variation of $f(d_{ij})$ is 0. 6020, which is the smallest when the travel distance of $d_{ij}$ is represented in kilometers. This illustrates that when $d_{ij}$ is expressed in kilometer units, the dispersion degree of $f(d_{ij})$ is the lowest, which is consistent with the coefficient of variation of travel distance expressed in kilometers.

In this experiment, we selected public transportation as the transportation mode for the STM G2SFCA model, the $\beta$ value being 1, and the $d_0$ being 100 km. In the MTM G2SFCA model and MTM-RTMC G2SFCA model, we selected four transportation modes, including walking, bicycling, public transportation, and driving, and the $d_0(m_r)$ were 60 km, 80 km, 100 km, and 120 km, respectively.

### 4.2. Comparison of the $V_{ij}$ Estimates with the Three G2SFCA Models

In the STM G2SFCA model, MTM G2SFCA model, and MTM-RTMC G2SFCA model, the influence factor $V_j$ gradually increases (Figure 9). The MTM-RTMC G2SFCA model shows significant improvement compared with the STM G2SFCA model and MTM G2SFCA model, indicating that RTMC significantly improves traffic accessibility. As traffic communication improves, $V_j$ increases, which means that each hospital will have a larger potential population.

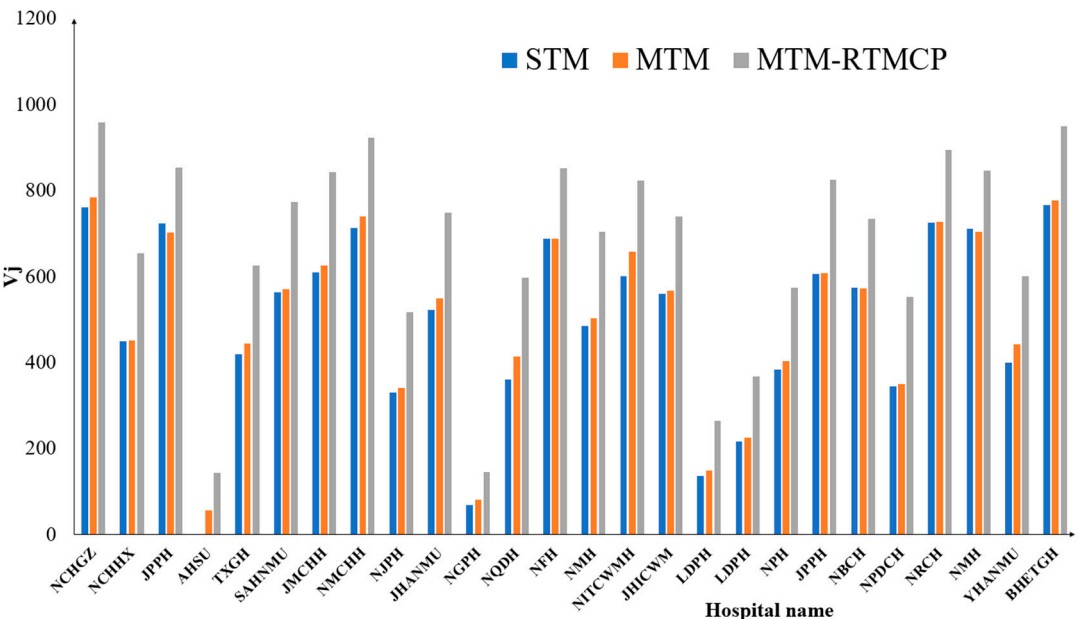

**Figure 9.** A variance bar chart of $V_j$.

### 4.3. Comparison of the Accessibility Estimates from the Three G2SFCA Models

A direct comparison of the overall research area (Figure 10a–c) suggests that the three models have the same spatial distribution characteristics, a binuclear aggregation distribution (for details, see supplementary materials). The phenomenon of binuclear aggregation distribution is mainly related

to the concentration of pediatric medical resources in NCHGZ and Nanjing Children's Hospital of Hexi Office (NCHHX): those attached to Nanjing Children's Hospital in Nanjing. Considering the number of pediatricians in each hospital, the number of pediatricians in Nanjing Children's Hospital consisting of NCHGZ and NCHHX is the largest, including 392 pediatricians, accounting for 65% of the total number of pediatricians in Nanjing. The maximum number of outpatient visits in Nanjing Children's Hospital exceeded 11,000, and the number of outpatient clinics in the evening was over 1000. The annual outpatient volume was 2,565,700. What is more, areas with a high accessibility value are located in the downtown, which owns the developed road network condition and multiple convenient transportation types. Good traffic conditions have been widely proven to be an essential factor in improving accessibility in existing research [3,76,77].

For the convenience of discussion, a series of local zoom maps were made for the core urban areas and external urban areas, as shown in Figure 10(a1,b1,c1,a2,b2,c2). For the main urban areas (Figure 10(a1,b1,c1)), the accessibility results of the MTM model and the MTM-RTCM model are significantly higher than those of the STM model, indicating that MTM is conducive to enlarging the coverage of high accessibility value. Because residents have more alternative modes of transportation, they tend to choose lower-cost modes. Additionally, there are some low-value areas in the core urban areas, mainly because these areas represent traffic barriers, such as water areas, green park space, undeveloped land, etc. In the external urban areas (Figure 10(a2,b2,c2)), there are significant differences in the radiated areas. However, they show the same level of accessibility values for the three models, shown as Figure 10(c2) > Figure 10(a2) > Figure 10(b2), indicating that the external urban areas also follow the basic principle of choosing a lower-cost mode of transportation. However, Figure 10(a2) > Figure 10(b2) shows that public transport plays a dominant role in the external urban areas, which is directly related to the traffic facilities and the spatial distance from medical facilities. The private car ownership rate in the external urban areas is lower than that in the core urban areas, and the spatial distance from medical facilities in the external urban areas is much higher than that in the core urban areas. These characteristics illustrate the significant influence of MTM-RTCM on accessibility.

The value of the STM G2SFCA model was significantly affected by the distribution conditions of the traffic network, which shows a distribution along with the main traffic network and with low accessibility overall. The global value of the STM G2SFCA model was 124.48. The value of the MTM G2SFCA model was presented based on concentric circle radiation, and the overall accessibility was dual-core. The global value of the MTM G2SFCA model was 129.67. The MTM-RTMC G2SFCA model offered concentric circular radiation, with a broader radiation range than MTM. The overall accessibility presented dual cores. The global value of the MTM-RTMC G2SFCA model was 132.53. Because the MTM-RTMC mechanism reduced the travel time of the overall area, the MTM-RTMC G2SFCA model improved accessibility and enlarged the accessibility radiation range. By choosing a more suitable transportation mode for their travel needs, residents can reduce the travel time to various healthcare facilities and improve the accessibility of the overall demand points.

To better reflect the changes in accessibility brought by the three models, the growth rates of the global value calculated by $\frac{A(mtm)-A(stm)}{A(stm)}*100\%$ and $\frac{A(mtm-rtmc)-A(mtm)}{A(mtm)}*100\%$ were increased by 4.1% and 2.2%, respectively. In the MTM-RTMC G2SFCA model, the closer the demand point was to the healthcare facilities, the more healthcare accessibility was weakened due to the low-speed transportation mode being prioritized. In contrast, the farther the demand point was from the healthcare facilities, the more significantly the healthcare accessibility of the MTM-RTMC G2SFCA model was enhanced compared to the STM G2SFCA model and the MTM G2SFCA model due to the improvement of transportation accessibility and convenience. The growth of accessibility has an immediate effect on the accessibility of the whole region. Examining the standard deviation in Table 3, the standard deviation of the MTM-RTMC G2SFCA model is the lowest, which signifies that the MTM-RTMC mechanism promotes the fairness of healthcare facilities and reduces the accessibility gap between demand points.

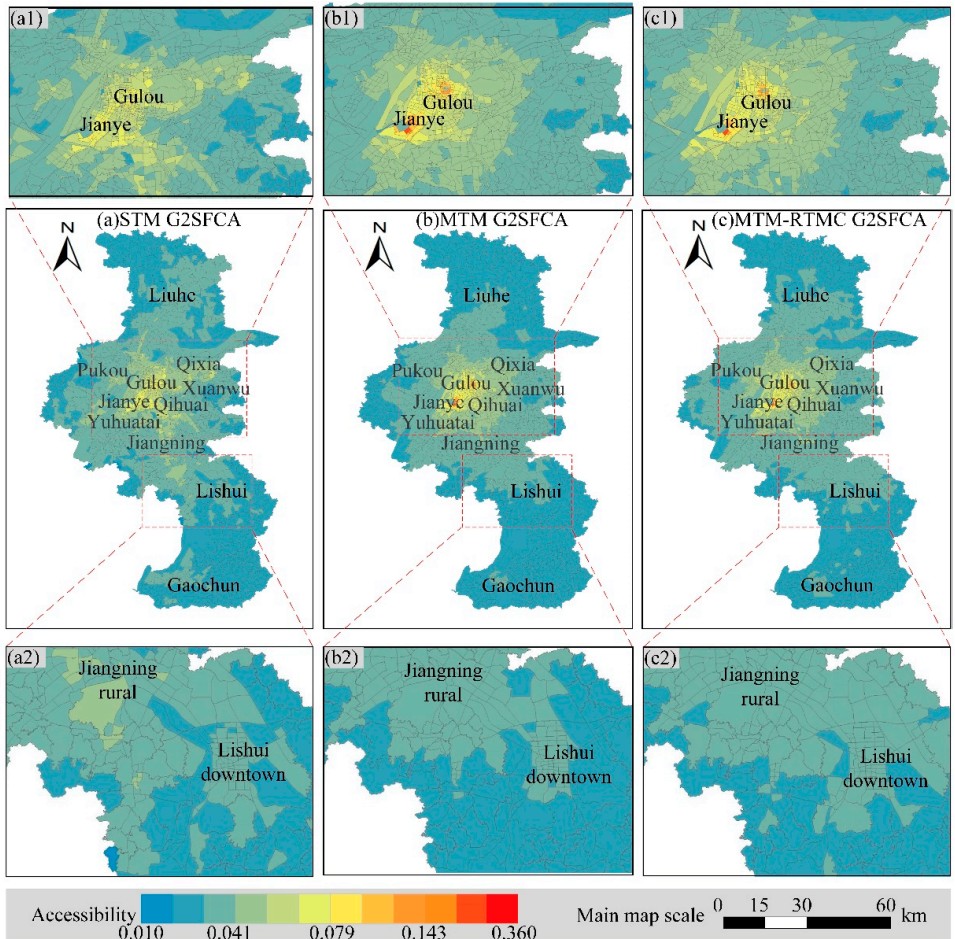

**Figure 10.** The accessibility results of the three G2SFCA models (STM, MTM, and MTM-RTMC) (high-resolution figure can be seen from https://figshare.com/s/7fc7a00e868a9c71ac37).

**Table 3.** The summary statistics of STM, MTM, and MTM-RTCM G2SFCA models' results.

|  | STM | MTM | MTM-RTCM |
|---|---|---|---|
| Minimum | 0 | 0.01243 | 0.00905 |
| Maximum | 0.42209 | 0.48843 | 0.36014 |
| Sum | 124.48 | 129.67 | 132.53 |
| Mean | 0.055 | 0.057 | 0.059 |
| Standard deviation | 0.049652 | 0.049644 | 0.041231 |

## 5. Discussion

### 5.1. The Potential Application of This MTM-RTCM Mechanism

A concern of this MTM-RTCM mechanism is its extension to the gravity model of spatial accessibility, of which 2SFCA is a special case [16,47]. Five major 2SFCA models include the generalized 2SFCA (base model) (Figure 11(a1)), enhanced 2SFCA (Figure 11(b1)), G2SFCA (Figure 11(c1)), kernel density 2SFCA (Figure 11(d1)), and Gaussian 2SFCA (Figure 11(e1)). Luo and Qi [77] proposed an enhanced 2SFCA method to identify more underserved areas by dividing a catchment into different weighted zones. Dai [78,79] integrated the 2SFCA method with a Gaussian function and kernel density function to measure accessibility while continuously discounting accessibility. Considering the strong subjectivity, the $d_{ij}$ catchment threshold classification standard of the generalized 2SFCA model and the enhanced 2SFCA model is difficult to determine. The parameter $\beta$ of the G2SFCA model establishes

subjectivity, while the kernel density 2SFCA model and Gaussian 2SFCA model calculation are clear, with no adjustment.

Following similar patterns from the MTM-RTCM mechanism, the MTM-RTMC mechanism can be generalized to five major 2SFCA models through improved distance decay functions (Table 4). The improved distance decay functions expand traditional single transportation types to multiple transportation types, which means that the mathematical morphology of the improved distance decay functions evolves into 3-dimensional space (Figure 11(a2,b2,c2,d2,e2)). The master equation of different 2SFCA models with the MTM-RTMC mechanism continues to keep pace with Equation (11).

### 5.2. Implications and Theoretical Thinking of the Proposed Method

The MTM-RTCM mechanism can offer pragmatic implications for policy initiatives regarding the influence of MTM-RTCM on new services allocation, especially in residential service planning policy. Traditional methods of residential service planning policy are based on the irrelative spatial unit to set new locations of service facilities. However, we assume a more reasonable way is that the whole city, as an organic whole, constructs dynamic zones of services based on the road network and MTM-RTMC, which will promote the services locations to be efficient and intensive throughout the city. Typical application directions include optimizing healthcare services location-allocation problems based on MTM-RTCM [53,80] and improving the spatial equity of multilevel healthcare in the metropolis based on MTM-RTCM [11].

**Table 4.** List of MTM-RTMC mechanisms improving the distance decay function of five 2SFCA models.

| Model Name | The Distance Decay Function | The Improved Distance Decay Function with MTM |
|---|---|---|
| Generalized 2SFCA | $f(d_{ij}) = \begin{cases} \theta_1, & d_{ij} \le d_0 \\ 0, & d_{ij} > d_0 \end{cases}$ | $f(d_{ij}(m_r)) = \begin{cases} \theta_1(m_r), & d_{ij}(m_r) \le d_0(m_r) \\ 0, & d_{ij}(m_r) > d_0(m_r) \end{cases}$ |
| Enhanced 2SFCA | $f(d_{ij}) = \begin{cases} \theta_1, & d_{ij} \in d_1 \\ \theta_2, & d_{ij} \in d_2 \\ \dots, & \dots \\ \theta_U, & d_{ij} \in d_U \end{cases}$ | $f(d_{ij}(m_r)) = \begin{cases} \theta_1(m_r), & d_{ij}(m_r) \in d_1(m_r) \\ \theta_2(m_r), & d_{ij}(m_r) \in d_2(m_r) \\ \dots, & \dots \\ \theta_U(m_r), & d_{ij}(m_r) \in d_U(m_r) \end{cases}$ |
| G2SFCA | $f(d_{ij}) = \begin{cases} d_{ij}^{-\beta}, & d_{ij} \le d_0 \\ 0, & d_{ij} > d_0 \end{cases}$ | $f(d_{ij}(m_r)) = \begin{cases} (d_{ij}(m_r))^{-\beta(m_r)}, & d_{ij}(m_r) \le d_0(m_r) \\ 0, & d_{ij}(m_r) > d_0(m_r) \end{cases}$ |
| Kernel Density 2SFCA | $f(d_{ij}) = \frac{3}{4} * \left[1 - \left(\frac{d_{ij}}{d_0}\right)^2\right], d_{ij} \le d_0$ | $f(d_{ij}(m_r)) = \frac{3}{4} * \left[1 - \left(\frac{d_{ij}(m_r)}{d_0(m_r)}\right)^2\right], d_{ij}(m_r) \le d_0(m_r)$ |
| Gaussian 2SFCA | $f(d_{ij}) = \frac{e^{(-1/2)*(d_{ij}/d_0)^2} - e^{-1/2}}{1 - e^{-1/2}}, d_{ij} \le d_0$ | $f(d_{ij}(m_r)) = \frac{e^{(-1/2)*(d_{ij}(m_r)/d_0(m_r))^2} - e^{-1/2}}{1 - e^{-1/2}}, d_{ij}(m_r) \le d_0(m_r)$ |

From the view of the method innovation perspective, our methodology shares the same basic design concept with Tahmasbi's et al. [22] research comparing the most similar and nearby multimodal accessibility studies. Tahmasbi et al. took into account the effects of different income groups in Isfahan, Iran, but the traffic travel times of different transportation modes were still based on the simulation calculation of GIS software, which failed to take into account the effects of various traffic factors on resident travel and preference. Such comparison intuitively shows where we have innovated and where we need to improve. From the view of the knowledge discovery of a PCS perspective, what motivates the phenomenon that pediatrics in Nanjing is showing an aggregate developmental effect? The two highest scored regions and three subcenters with relatively high accessibility values are

in Nanjing. The phenomenon is also different from the patient-centered medical home, an approach to providing comprehensive primary care for children and youth in developed countries, such as the US and UK [81]. The deeper reason behind this phenomenon that we suspect should be directly related to the complexity of pediatric specialties, the difficulty of pediatrician training, and sensitivity to pediatric treatment [71]. We hold an opinion that medical collectivization [82], high-level hospitals in the region collaborating with other health services, and community-based primary care may be conducive to the PCS development of quality and equity.

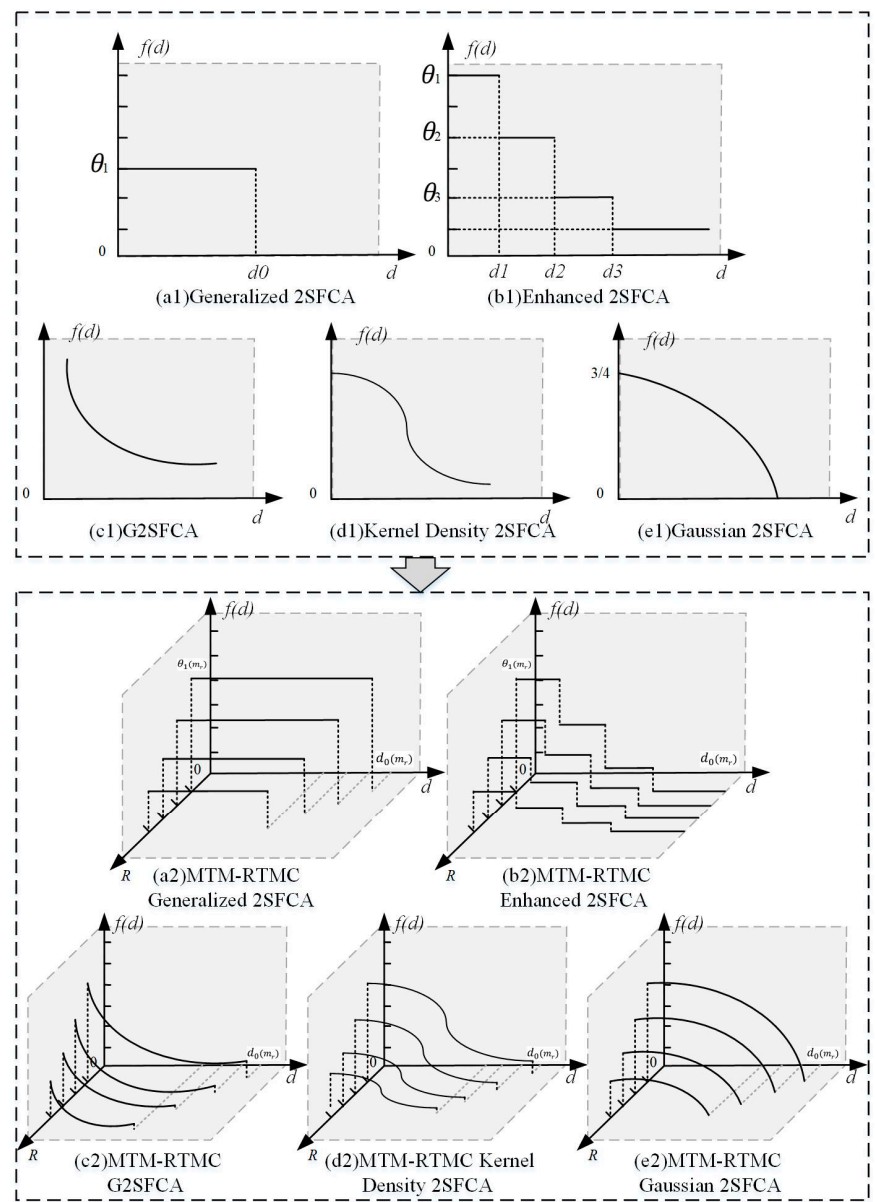

**Figure 11.** Different distance decay functions.

## 6. Conclusions

In this paper, we attempted to fill a gap in the knowledge of MTM-RTMC by presenting the accessibility influence of PCS for Nanjing, one of the Chinese megacities. We integrated the MTM-RTMC mechanism into the G2SFCA model and generalized it to five significant types of 2SFCA models. The MTM-RTMC G2SFCA model has a considerable effect on the traditional estimation of travel impedance, which depends on whether the MTM-RTMC mechanism enriches a single transportation

mode to multiple transportation modes of inhabitants. The MTM-RTMC mechanism provides a more reliable way than the STM to measure accessibility between demand locations to provider locations by the choice mechanism of multiple transportation modes, especially in far distant places, such as suburbs and the countryside. The MTM-RTMC mechanism, which works in dynamic traffic environments in modern cities, can be helpful for policymakers to measure the accessibility of service provision.

The MTM-RTMC mechanism, which has been carried out on the preliminary proof, is proposed to improve the accessibility calculation to portray real residents' travel characteristics more accurately. However, there are still research questions to be addressed. How can RTMC probability differences be characterized by comprehensive factors, such as transportation service facilities provision, weather conditions, seasonal conditions, regional topography features, commuting timetables, and regional transportation characteristics (for example, Hanoi in Vietnam is a city where the main form of transportation is the motorcycle)? It is a challenge for future related research. We believe that more factors can be considered in combination with the research method applied through the humanistic geography questionnaire. We will commence experimental analysis and comparison of five major 2SFCA models with the MTM-RTMC mechanism in future research. More importantly, based on the government's transportation big data platform that can collect daily travel data of individuals, we will verify the consistency between the estimated MTM-RTMC probabilities and real travelers' preferences of different transport modes.

**Supplementary Materials:** The data supporting the findings are available in "figshare.com" with the private link https://figshare.com/s/185e7f9682f791527cdc. We have provided the code, including that initializing the list of OD flows, requesting real-time route planning time and distance from Amap maps, and calculating the accessibility results of the three models. The code can be run in Python IDE (such as "PyCharm 2017"). The research data, including "rtmcp_mtm_gm.json" data and "street_block_mock_data.shp" data, provided the necessary data for accessibility calculations and spatial symbolization. Considering the state secrecy of the street block data, we used the ArcGIS buffer method and ArcGIS feature envelope to polygon method to generate mock data consistent with the street block data morphology and properties. The "rtmcp_mtm_gm.json" data can be opened using the software "MongoDB 3.2," and the "street_block_mock_data.shp" data can be opened using the software "ArcGIS 10.2." All of those research data were collected from web open-source approaches and did not involve any personally sensitive information, except the street block shapefile. So, the ethical approval is in low risk.

**Author Contributions:** Conceptualization, Changbin Wu; methodology, Xinxin Zhou; software, Xinxin Zhou; validation, Lei Wang and Zhaoyuan Yu; formal analysis, Xinxin Zhou; investigation, Linwang Yuan and Changbin Wu; data curation, Xinxin Zhou and Changbin Wu; writing—original draft preparation, Xinxin Zhou; writing—review and editing, Xinxin Zhou and Zhaoyuan Yu; visualization, Xinxin Zhou and Zhaoyuan Yu; supervision, Changbin Wu; project administration, Changbin Wu; funding acquisition, Changbin Wu. All authors have read and agreed to the published version of the manuscript.

**Funding:** This research was supported by China National Funds for NSFC, grant number 41471318; China National Funds for Distinguished Young Scientists, grant number 41625004; and Research and Innovation Plan for Graduate Students in Jiangsu Province in 2020, grant number KYCX20_1183.

**Acknowledgments:** The authors would like to thank the anonymous referees and editor for their valuable comments, which significantly improved this article.

**Conflicts of Interest:** The authors declare no conflict of interest.

## Appendix A

**Table A1.** Hospitals in Nanjing offering pediatric services.

| ID | Hospital Name | Abbreviation | Longitude | Latitude | Number of Pediatricians | Hospital Level |
|---|---|---|---|---|---|---|
| 1 | Nanjing Children's Hospital Guangzhou Road Office | NCHGZ | 118.7742 | 32.0527 | 157 | 3 |
| 2 | Nanjing Children's Hospital Hexi Office | NCHHX | 118.7026 | 31.9840 | 235 | 3 |
| 3 | Jiangsu Provincial People's Hospital | JPPH | 118.7600 | 32.0500 | 26 | 3 |
| 4 | Affiliated Hospital of Southeast University | AHSU | 118.7720 | 32.7020 | 13 | 3 |

**Table A1.** *Cont.*

| ID | Hospital Name | Abbreviation | Longitude | Latitude | Number of Pediatricians | Hospital Level |
|----|---------------|--------------|-----------|----------|-------------------------|----------------|
| 5 | Taikang Xianlin Gulou Hospital | TXGH | 118.9330 | 32.0950 | 6 | 3 |
| 6 | Second Affiliated Hospital of Nanjing Medical University | SAHNMU | 118.7390 | 32.0810 | 26 | 3 |
| 7 | Jiangsu Maternal and Child Healthcare Hospital | JMCHH | 118.7360 | 32.0600 | 23 | 3 |
| 8 | Nanjing Maternal and Child Healthcare Hospital | NMCHH | 118.7710 | 32.0410 | 14 | 3 |
| 9 | Nanjing Jiangbei People's Hospital | NJPH | 118.7520 | 32.2370 | 3 | 3 |
| 10 | Jiangning Hospital affiliated to Nanjing Medical University | JHANMU | 118.8440 | 31.9500 | 8 | 3 |
| 11 | Nanjing Gaochun People's Hospital | NGPH | 118.8650 | 31.3220 | 20 | 2 |
| 12 | Nanjing Qixia District Hospital | NQDH | 118.8820 | 32.1220 | 2 | 2 |
| 13 | Nanjing First Hospital | NFH | 118.7870 | 32.0220 | 8 | 3 |
| 14 | Nanjing Mingji Hospital | NMH | 118.7200 | 31.9810 | 6 | 3 |
| 15 | Nanjing Integrated Traditional Chinese and Western Medicine Hospital | NITCWMH | 118.8530 | 32.0350 | 9 | 3 |
| 16 | Jiangsu Hospital of Integrated Chinese and Western Medicine | JHICWM | 118.8040 | 32.0990 | 6 | 3 |
| 17 | Lishui District People's Hospital | LDPH | 119.0270 | 31.6320 | 6 | 3 |
| 18 | Liuhe District People's Hospital | LDPH | 118.8400 | 32.3420 | 9 | 2 |
| 19 | Nanjing Pukou Hospital | NPH | 118.7140 | 32.1060 | 4 | 2 |
| 20 | Jiangsu Province Provincial Hospital | JPPH | 118.7360 | 32.0670 | 2 | 3 |
| 21 | Nanjing Branch of Changzheng Hospital | NBCH | 118.7440 | 32.0930 | 1 | 3 |
| 22 | Nanjing Pukou District Central Hospital | NPDCH | 118.6320 | 32.0500 | 3 | 2 |
| 23 | Nanjing Red Cross Hospital | NRCH | 118.7870 | 32.0280 | 1 | 2 |
| 24 | Nanjing Municipal Hospital | NMH | 118.7890 | 32.0560 | 1 | 2 |
| 25 | Yifu Hospital Affiliated to Nanjing Medical University | YHANMU | 118.8880 | 31.9330 | 8 | 3 |
| 26 | Bayi Hospital of Eastern Theater General Hospital | BHETGH | 118.7880 | 32.0360 | 6 | 3 |

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
