# Peer review of "Measuring Accessibility of Healthcare Facilities for Populations with Multiple Transportation Modes Considering Residential Transportation Mode Choice"

_ijgi, doi:10.3390/ijgi9060394_

Round 1

Reviewer 1 Report

Review Report

Date: 21/05/2020

Yu et. al. conducted a study to measure accessibility of healthcare facilities for populations with multiple transportation mode considering residential transportation mode choice. In my opinion, this study is relevant but has some weaknesses which should be addressed before consideration for publication.

Abstract

The results section of the abstract lack some relevant information. Please include the different transport choices available in the study area.

What is the estimated travel impedance? Mean (SD) in Kilometers.

Which transportation choice offers the best accessibility?

Method and results

Four transportation modes (driving, walking, public transportation, bicycling) were considered yet the results did show the travel time variation for each of these modes. This information will essential.

Information on the correlation between travel time and the utilization of the service will also be relevant.

A review was conducted as part of this study yet there is no description of the review method. Please include this.

The first item in Table 1 does not have the author information.

Please indicate the year of publication in Table 1. E.g Yu et. al., 2020

Did you check the accuracy of the assessment carried out through Spatial Analysis (GIS)? I think this will be necessary

Discussion

Please include the following:

Summary of your study findings

Compare your study findings with studies conducted in other jurisdictions

The strengths and limitations of your study

The implications of your study findings, etc

Ethical approval

There was no mention of ethical approval for this study. Can you explain why?

I hope this helps.

Thank you

Author Response

Response to Reviewer 1 Thanks for your comments on our paper. We have revised our paper according to your comments. Yu et. al. conducted a study to measure accessibility of healthcare facilities for populations with multiple transportation mode considering residential transportation mode choice. In my opinion, this study is relevant but has some weaknesses which should be addressed before consideration for publication. Abstract Comment 1. The results section of the abstract lack some relevant information. Please include the different transport choices available in the study area. Answer. Thank you very much for the excellent and professional review for this manuscript. In this revised version, changes to the results section of the abstract were all highlighted within the document. Comment 2. What is the estimated travel impedance? Mean (SD) in Kilometers. Answer. The travel impedance mainly refers to the coefficient of travel difficulty between provider locations to demand locations, including travel time, distance (Mean (SD) in Kilometers), or travel expense. Travel impedance has been widely proven by scholars to be a key factor affecting residents' access to service facilities [1]. We also uniformed the terms, such as traffic impedance. Comment 3. Which transportation choice offers the best accessibility? Answer. The problem mainly varies with provider locations and demand locations and also changes with kinds of travel impact factors. Due to differences in travel impact factors such as traffic conditions, origin location, distance to the destination, and economic cost, residents' daily travel presents different residential transportation mode choices (RTMC). We also elaborated and emphasized this in the manuscript. Method and results Comment 4. Four transportation modes (driving, walking, public transportation, bicycling) were considered yet the results did show the travel time variation for each of these modes. This information will essential. Answer. According to your comments, we have revised the manuscript extensively in Fig. 7. For ease of expression, we select an empirical provider location, NCHGZ, to show the travel time and distance from different street blocks requested from the route planning API of Amap Maps (point-in-time: October 8, 2019)(Fig. 7). Fig 7 (a1, a2, a3, a4) presents that the travel distance of different transportation modes is different to NCHGZ, which results from that different transportation modes choose kinds of trip routes. The travel time variability of different transportation modes has apparent differences shown in Fig 7 (b1, b2, b3, b4). The most variability and fastest way is in driving, followed by public transportation while walking and bicycling have evolved as a series of concentric rings. Comment 5. Information on the correlation between travel time and the utilization of the service will also be relevant. Answer. Yes, the utilization of the service also is one of the factors for accessibility research [2,3]. Thank you for your helpful comments on our article. This is not the focus of this paper's scientific question, so we don't carry out in this paper. But we will study this in the future when conditions permit. Comment 6. A review was conducted as part of this study yet there is no description of the review method. Please include this. Answer. Thank you for your reminding. We have made significant adjustments in the introduction from three aspects, including multimodal transportation, constant theoretical value, and residential transportation mode choices, to highlight the research venation. At the same time, in Section 2, we also made a classification review of accessibility methods. Comment 7. The first item in Table 1 does not have the author information. Please indicate the year of publication in Table 1. E.g Yu et. al., 2020. Answer. In this revised version, we selected two empirical OD flows, Yujinli Community to Nanjing Children Hospital of Guangzhou Office (NCHGZ) (Fig. 5(a)) and Yujinli Community to Yifu Hospital Affiliated to Nanjing Medical University (YHANMU) (Fig. 5(b)), to obtain the recommend paths of MTM from Amap Maps (point-in-time: October 8, 2019). Comment 8. Did you check the accuracy of the assessment carried out through Spatial Analysis (GIS)? I think this will be necessary Answer. All experiments and mapping expressions in this paper are realized based on the most popular GIS platform, ArcGIS. Discussion Comment 9. Please include the following: (1) Summary of your study findings; (2) Compare your study findings with studies conducted in other jurisdictions; (3) The strengths and limitations of your study; (4) The implications of your study findings, etc. Answer. We feel great thanks for your professional review work on our article. As you are concerned, there are several emphasizes that need to be addressed. The MTM-RTMC G2SFCA model has a considerable effect on the traditional estimation of travel impedance, which depends on that the MTM-RTMC mechanisms enrich single transportation mode to multiple transportation modes of inhabitants. The MTM-RTMC G2SFCA model provides a more reliable way to measure accessibility between demand locations to provider locations by the choice mechanism of multiple transportation modes, especially in far distant places, such as suburbs and the countryside. We attempt to fill a gap in the knowledge of MTM-RTMC by presenting the accessibility influence of pediatric clinic services for Nanjing, one of the Chinese megacities. Ethical approval Comment 10. There was no mention of ethical approval for this study. Can you explain why? Answer. All of those research data are collected from open-source web approaches and are not involving any personally sensitive information, except the street block shapefile. So the ethical approval is at low risk.

Reviewer 2 Report

The paper aims at proposing a new methodology to measure accessibility. Relying on pediatric hospitals in Nanjing, it is evaluated how the MTM-RTMC solution provides a more realistic measurement of accessibility.

Major comments:

  • I would suggest the authors to separate literature on MTM and RTMC from the introduction. The introduction should focus more on explaining why the topic is of interest and which are the novelties that this paper presents, without providing too many details on the technical part. Additionally, it would be interesting to read a paragraph evidencing why accessibility to healthcare facilities is a topic of great interest and underlining the reason why Nanjing is a good testing bed.
  • Please specify better which is the purpose of the paper, the assumptions that the author take and the final results. There are a lot of compared models and the reader sometimes has difficulties in understand which is the “novelty” presented in the work.
  • I really appreciate the illustration of the RTMC factors affecting travel costs. I would suggest the authors to link the different identified factors with previous work from the literature and to better discuss the existing relationships among these factors and travel cost.
  • Please explain better how the authors derive Table 3, specifically referring on what authors mean in lines 415-419.
  • Please explain better what the authors mean in lines 431-432.
  • I would suggest to spend more time in discussing the results listed in the conclusion, non in terms of numbers but in terms of implications. How do you link your higher variation in the accessibility value with respect A(stm) and A(mtm) with a better model? What are your suggestions with respect to travel distance decay functions and all the related terms?
  • What do the authors mean when they say that MTM-RTMC “improves spatial accessibility to medical facilities.”? Or that the “model can improve the accessibility and convenience of residents' transportation?”. It is not something that physically impact on accessibility, rather something for measuring it.

Other comments:

  • It is not clear how authors compute the hospital level. Does it depend on the number of pediatricians?
  • Please make explicit what are W B PT and D in Table 3
  • Please consider to modify all tables’ titles to make them self-explaining (e.g,. in Table 4 – minimum, maximum …, of what?)
  • I would suggest to remove Table 2 and substitute it with a map, where the number of pediatricians represent the size of the geographical point where the hospital is located. If authors think that is important to have the names of hospitals included in the paper, I would suggest to move that table in the Appendix
  • Where does the average speed per transport mode come from?
  • Point 3 is reported twice at page 7
  • Please consider to revise Fig.2 as it is difficult to link the legend colors with the graph (e.g., try to mark different points with a symbol)
  • Please unify the way in which formulas are presented. Sometimes the description of terms is on the top of them, sometimes at the bottom.
  • Please uniform the way in which papers are cited. (see line 413)
  • Please check for double spacing
  • The paper needs proofreading.

Author Response

Response to Reviewer 2 Major comments: Comment 1. I would suggest the authors to separate literature on MTM and RTMC from the introduction. The introduction should focus more on explaining why the topic is of interest and which are the novelties that this paper presents, without providing too many details on the technical part. Additionally, it would be interesting to read a paragraph evidencing why accessibility to healthcare facilities is a topic of great interest and underlining the reason why Nanjing is a good testing bed. Answer. We feel great thanks for your professional review work on our article. As you are concerned, We have made significant adjustments in the introduction from three aspects, including multimodal transportation, constant theoretical value, and residential transportation mode choices, to highlight the context of novelties that this paper presents. And we have cut short on the technical part. Due to space limitations, I hope the behavior that we have not separate literature on MTM and RTMC from the introduction will get your understanding. Although the average number of pediatricians per 1,000 people in Nanjing, being 0.67, is higher than the average for China, it is still far below the number for the principal developed countries, whose ratio reaches a standard of 0.85-1.3, which shows that a significant gap of pediatric resources. Therefore, considering the population, transportation, and medical services factors, Nanjing is selected as a metropolitan research area with strong typicality (Line 325-329). Comment 2. Please specify better which is the purpose of the paper, the assumptions that the author take and the final results. There are a lot of compared models and the reader sometimes has difficulties in understand which is the "novelty" presented in the work. Answer. We refined and summarized the purpose and results again. In this paper, The purpose of our study is to measure the spatial accessibility of healthcare facilities based on MTM considering RTMC (MTM-RTMC). We integrated the MTM-RTMC mechanism into the G2SFCA model and generalized it to five significant types of 2SFCA models. The MTM-RTMC G2SFCA model has a considerable effect on the traditional estimation of travel impedance, which depends on that the MTM-RTMC mechanisms enrich single transportation mode to multiple transportation modes of inhabitants. The MTM-RTMC G2SFCA model provides a more reliable way to measure accessibility between demand locations to provider locations by the choice mechanism of multiple transportation modes, especially in far distant places, such as suburbs and the countryside. We attempt to fill a gap in the knowledge of MTM-RTMC by presenting the accessibility influence of pediatric clinic services for Nanjing, one of Chinese megacities. The results show that the MTM-RTMC mechanism is to make up for the traditional estimation of accessibility, which loses sight of the influence of residential transportation choices. Comment 3. I really appreciate the illustration of the RTMC factors affecting travel costs. I would suggest the authors to link the different identified factors with previous work from the literature and to better discuss the existing relationships among these factors and travel cost. Answer. Thank you for your recognition. As we knew, a multitude of factors, including travel experience [4], travel purpose [5], real-time traffic information [5,6], departure time urgency [7], travel time and travel distance [8] would all impact residents' multimodal choice preferences for different transport modes. Comment 4. Please explain better how the authors derive Table 3, specifically referring on what authors mean in lines 415-419. Answer. Thank you for your reminding. We select public transportation as the transportation mode for the STM G2SFCA model, the β value being 1, and the d_0 being 100 km. In the MTM G2SFCA model and MTM-RTMC G2SFCA model, we select four transportation modes, including walking, bicycling, public transportation and driving, and the d_0 (m_r) are 60 km, 80 km, 100 km, and 120 km, respectively. Comment 5. Please explain better what the authors mean in lines 431-432. Answer. Thank you for your reminding. According to the reviewer's comments, we have explained the mean in lines 431-432 extensively. A direct comparison of the overall research area (Fig. 10 (a), (b), (c)) suggests that the three models have the same spatial distribution characteristics, a binuclear aggregation distribution. The phenomenon of binuclear aggregation distribution is mainly related to the concentration of pediatric medical resources in NCHGZ and NCHHX, which attach to Nanjing Children Hospital in Nanjing. Considering the number of pediatricians in each hospital, the number of pediatricians in Nanjing Children Hospital consisting of NCHGZ and Nanjing Children Hospital of Hexi Office (NCHHX) is the largest, including 392 pediatricians, accounting for 65% of the total number of pediatricians in Nanjing. The maximum number of outpatient visits in Nanjing Children Hospital exceeded 11,000, and the number of outpatient clinics in the evening was over 1,000. The annual outpatient volume was 2,565,700. What's more, areas with high accessibility value are located in the downtown, which owns the developed road network condition and multiple convenient transportation types. Right traffic conditions have been widely proven to be an essential factor in improving accessibility in existing research [9-11]. Comment 6. I would suggest to spend more time in discussing the results listed in the conclusion, non in terms of numbers but in terms of implications. How do you link your higher variation in the accessibility value with respect A(stm) and A(mtm) with a better model? What are your suggestions with respect to travel distance decay functions and all the related terms? Answer. According to the reviewer's comments, we have revised the manuscript extensively. And we have cut short on the terms of numbers part. Comment 7. What do the authors mean when they say that MTM-RTMC "improves spatial accessibility to medical facilities."? Or that the "model can improve the accessibility and convenience of residents' transportation?". It is not something that physically impacts on accessibility, rather something for measuring it. Answer. The expression in the original manuscript about "MTM-RTMC improves spatial accessibility to medical facilities" has a certain context of ambiguous. We unified the summary expression of MTM results. The MTM-RTMC mechanism is to make up for the traditional estimation of accessibility, which loses sight of the influence of residential transportation choices. The MTM-RTMC mechanism that provides a more realistic and reliable way can generalize to major accessibility models and offers preferable guidance for policymakers. And we hope the revised manuscript could be acceptable for you. Other comments: Comment 8. It is not clear how authors compute the hospital level. Does it depend on the number of pediatricians? Answer. According to the reviewer's comments, we have revised the manuscript extensively. Comment 9. Please make explicit what are W B PT and D in Table 3 Answer. We have revised the manuscript extensively. Comment 10. Please consider to modify all tables' titles to make them self-explaining (e.g,. in Table 4 – minimum, maximum …, of what?) Answer. We have revised the manuscript. Comment 11. I would suggest to remove Table 2 and substitute it with a map, where the number of pediatricians represent the size of the geographical point where the hospital is located. If authors think that is important to have the names of hospitals included in the paper, I would suggest to move that table in the Appendix Answer. We have revised the manuscript. Comment 12. Where does the average speed per transport mode come from? Answer. We have revised the manuscript. Comment 13. Point 3 is reported twice at page 7 Answer. It was indeed an organization error. We have revised the manuscript. Comment 14. Please consider to revise Fig.2 as it is difficult to link the legend colors with the graph (e.g., try to mark different points with a symbol) Answer. We have revised the manuscript. Comment 15. Please unify the way in which formulas are presented. Sometimes the description of terms is on the top of them, sometimes at the bottom. Answer. We have made the uniform the formulas to the bottom within the whole manuscript. Comment 16. Please uniform the way in which papers are cited. (see line 413) Answer. We have made the uniform the way in which papers are cited. Comment 17. Please check for double spacing Answer. Based on your comments, we have made the corrections of double-spacing within the whole manuscript. Comment 18. The paper needs proofreading. Answer. The manuscript has undergone English proofreading.

Reviewer 3 Report

The paper discusses the application of different methods for assessing accessibility index. it focuses mainly on the analysis of multimodal accessibility and travel friction coefficient.

Sometimes during the reading there is confusion between assumptions and results. Even if the method suggested is theoretically able to better measures accessibility, at the end of the application to the case study analyzed (pediatric hospital accessibility in Nanjing) authors fail in convincing the reader about the advantage of the method proposed. probably because the paper does not provide evidence that this new accessibility index better represents the disutility for citizens (for example the ability to better explain their choices).The authors state that "MTM is conducive to improving accessibility" I do not understand the meaning. The fact that MTM suggest higher values does not imply that the index is more adequate.

the second point regards the administrative division employed and its effect on the analysis. The densely populated areas are divided into smaller sub-unit ( in terms of extension) while non-urban areas can be quite large. this means that large areas are associated to a specific access time ( multiple for MTM) to each hospital. is it correct? , i.e the unique distance value (let say by car) associated with a specific hospital to an area of 100 square kilometres is really a representative value for all inhabitants of that cell?

what happen to the results if we switch to a homogeneous grid?

Last point: It is not clear what are the final guidance suggested by the authors for the choice of the beta parameter

Author Response

Comment 1. The paper discusses the application of different methods for assessing accessibility index. it focuses mainly on the analysis of multimodal accessibility and travel friction coefficient. Sometimes during the reading there is confusion between assumptions and results. Even if the method suggested is theoretically able to better measures accessibility, at the end of the application to the case study analyzed (pediatric hospital accessibility in Nanjing) authors fail in convincing the reader about the advantage of the method proposed. probably because the paper does not provide evidence that this new accessibility index better represents the disutility for citizens (for example the ability to better explain their choices). The authors state that "MTM is conducive to improving accessibility" I do not understand the meaning. The fact that MTM suggest higher values does not imply that the index is more adequate.

Answer. Thank you for your reminding. You and the other two reviewers' comments are all of great importance to our article. The expression in the original manuscript about "MTM-RTMC improves spatial accessibility to medical facilities" has a certain context of ambiguous. We unified the summary expression of MTM results. The MTM-RTMC mechanism is to make up for the traditional estimation of accessibility, which loses sight of the influence of residential transportation choices. The MTM-RTMC mechanism that provides a more realistic and reliable way can generalize to major accessibility models and offers preferable guidance for policymakers. And we hope the revised manuscript could be acceptable for you.

Meanwhile, we also wrote a point-by-point response letter to the other two nice reviewers to acknowledge their help and denote where we made revisions. In this paper, we integrated the MTM-RTMC mechanism into the G2SFCA model and generalized it to five significant types of 2SFCA models. The MTM-RTMC G2SFCA model has a considerable effect on the traditional estimation of travel impedance, which depends on that the MTM-RTMC mechanisms enrich single transportation mode to multiple transportation modes of inhabitants and provide a more reliable way to measure accessibility between demand locations to provider locations by choice mechanism of multiple transportation modes, especially in far distant places, such as suburbs and the countryside. We attempt to fill a gap in the knowledge of MTM-RTMC by presenting the accessibility influence of pediatric clinic services for Nanjing, one of Chinese megacities. The results show that the MTM-RTMC mechanism is to make up for the traditional estimation of accessibility which loses sight of the influence of residential transportation choices. The MTM-RTMC mechanism that provides a more realistic and reliable way of measuring citizens' trips can generalize to major accessibility models and offers preferable guidance for policymakers.

Comment 2. The second point regards the administrative division employed and its effect on the analysis. The densely populated areas are divided into smaller sub-unit ( in terms of extension) while non-urban areas can be quite large. this means that large areas are associated to a specific access time (multiple for MTM) to each hospital. is it correct? , i.e the unique distance value (let say by car) associated with a specific hospital to an area of 100 square kilometres is really a representative value for all inhabitants of that cell? What happen to the results if we switch to a homogeneous grid?

Answer. This is a good suggestion. The selection of reasonable spatial statistical units has always been one of the critical factors affecting geographic science modeling. In the study of spatiotemporal prediction, the spatial statistical unit can be divided into administrative division based (irregular region), traffic analysis based (irregular region), geographic grid-based (regular region), and network-based [12]. Different statistical units have a great influence on the expression of the predicted results, which also exists in accessibility studies, modifiable areal unit problem[13,14]. The modifiable areal unit problem would lead to the difference in accessibility evaluation results [13], which depends on the influence of multiple scales of spatial statistical units. The street block data are the spatial framework data for realizing unified cadastral management, and their division will be comprehensively considered based on organizational factors, natural geographical conditions, and public facilities [15]. The street blocks in central urban areas demonstrate small and dense areas, with some being smaller than 1 km2. In contrast, the street blocks in peripheral urban areas are large areas and sparsely populated, with some being more than 20 km2. Therefore, using street block data as a research unit can better characterize urban morphological characteristics.

We also used the grid-based method to experiment (Fig. 1), and its shortcomings were mainly concentrated in the main urban areas, where the 1KM*1KM grid was too large. So that many detailed features could not be displayed, while for the suburbs, the 1KM*1KM grid was too small, which increased many invalid calculations.

Fig 1. the experiment result of the grid-based method

Comment 3. Last point: It is not clear what are the final guidance suggested by the authors for the choice of the beta parameter.

Answer. Therefore, in the MTM G2SFCA model and MTM-RTMC G2SFCA model, we assign the travel friction coefficients as 1.4, 1.2, 1.0, 0.8 for walking, bicycling, public transportation and driving, respectively, which conforms to the suppressing effect on the speed of transportation modes.

Reviewer 4 Report

In this paper the authors propose a new method to measure spatial accessibility, focusing on the accessibility of healthcare facilities in the city of Nanjing as an example. In particular, they build upon the G2SFCA method by incorporating both multiple modes of transport and the likelihood that residents will choose that mode.

I believe that the proposed methodology is clear and relevant to this journal. However, there are a few things that should be addressed before publication:

- The example given in line 53 regarding the possibility of choosing multiple routes or modes of transport is too specific for a broad audience (going from Shoukai square to Wangjing SOHO). Besides, the citation is just the main page of the website, which does not include any information about those two places. I would rather suggest citing some of the literature focused on route planning and the interplay of different modes of transport such as:

Gallotti, R., Barthelemy, M. The multilayer temporal network of public transport in Great Britain. Sci Data 2, 140056 (2015). https://doi.org/10.1038/sdata.2014.56
Gallotti, R., Porter, M. and Barthelemy, M. Lost in transportation: Information measures and cognitive limits in multilayer navigation. Science Advances 2, e1500445 (2016). https://doi.org/10.1126/sciadv.1500445
Aleta, A., Meloni, S. & Moreno, Y. A Multilayer perspective for the analysis of urban transportation systems. Sci Rep 7, 44359 (2017). https://doi.org/10.1038/srep44359

- I think that there is a mistake in equation 10 since adding w(m_r) essentially turns the operation into a weighted mean. If the weights of each mode r are correctly normalized (which I assume since it is said that sum_r w_ij(m_r) = 1), then it is not necessary to divide by R anymore. This is correctly taken into account in equation 11 but not in 10.

- The last sentence in line 275 has to be rewritten. The mean cannot be the standard deviation of the data. I understand the meaning (mu is the mean, sigma is the standard deviation) but it should be clarified.

- The sentence in lines 280-281 is a bit confusing. I guess that the authors wanted to give some examples of where the Softmax function can be used. Since it is not crucial for the understanding of the step, I would either remove or, at least, rewrite it.

- I do not understand the reference in line 345 to Google Maps API since in line 337 it is said that the paper uses Amap Maps. This should be clarified.

I have also found some minor spelling mistakes:

Page 5, line 170 model -> mode
Page 9, line 269 the larger is -> the larger it is

Author Response

Response to Reviewer 4

Comment 1. In this paper the authors propose a new method to measure spatial accessibility, focusing on the accessibility of healthcare facilities in the city of Nanjing as an example. In particular, they build upon the G2SFCA method by incorporating both multiple modes of transport and the likelihood that residents will choose that mode.

I believe that the proposed methodology is clear and relevant to this journal. However, there are a few things that should be addressed before publication:

- The example given in line 53 regarding the possibility of choosing multiple routes or modes of transport is too specific for a broad audience (going from Shoukai square to Wangjing SOHO). Besides, the citation is just the main page of the website, which does not include any information about those two places. I would rather suggest citing some of the literature focused on route planning and the interplay of different modes of transport such as:

Gallotti, R., Barthelemy, M. The multilayer temporal network of public transport in Great Britain. Sci Data 2, 140056 (2015). https://doi.org/10.1038/sdata.2014.56
Gallotti, R., Porter, M. and Barthelemy, M. Lost in transportation: Information measures and cognitive limits in multilayer navigation. Science Advances 2, e1500445 (2016). https://doi.org/10.1126/sciadv.1500445 
Aleta, A., Meloni, S. & Moreno, Y. A Multilayer perspective for the analysis of urban transportation systems. Sci Rep 7, 44359 (2017). https://doi.org/10.1038/srep44359

Answer. According to your suggestion, we have corrected this part into what you suggest.

Comment 2. I think that there is a mistake in equation 10 since adding w(m_r) essentially turns the operation into a weighted mean. If the weights of each mode r are correctly normalized (which I assume since it is said that sum_r w_ij(m_r) = 1), then it is not necessary to divide by R anymore. This is correctly taken into account in equation 11 but not in 10.

Answer. I'm terribly sorry about this small mistake. Also thank you very much for your careful review of this article. According to your suggestion, we have corrected this part into what you suggest.

Comment 3. The last sentence in line 275 has to be rewritten. The mean cannot be the standard deviation of the data. I understand the meaning (mu is the mean, sigma is the standard deviation) but it should be clarified.

Answer. According to your suggestion, we have corrected this part into what you suggest.

Comment 4. The sentence in lines 280-281 is a bit confusing. I guess that the authors wanted to give some examples of where the Softmax function can be used. Since it is not crucial for the understanding of the step, I would either remove or, at least, rewrite it.

Answer. According to your suggestion, we have corrected this part into what you suggest.

Comment 5. I do not understand the reference in line 345 to Google Maps API since in line 337 it is said that the paper uses Amap Maps. This should be clarified.

Answer. I'm sorry about this small mistake. According to your suggestion, we have corrected this part into what you suggest.

Comment 6. I have also found some minor spelling mistakes:

Page 5, line 170 model -> mode
Page 9, line 269 the larger is -> the larger it is

Answer. Thank you very much for the excellent and professional review for this manuscript. It was indeed a serious grammatical error. According to your suggestion, we have corrected this part into what you suggest.

Round 2

Reviewer 1 Report

I am satisfied with the Authors' responses to my comments except those at the discussion section. 

  1. The first paragraph of the discussion section looks awkward. There are about four figures and one table within one sentence. I think that is inappropriate in my opinion. 
  2. I still think the authors should discuss their study findings in comparison with similar studies conducted in other jurisdictions/countries. My previous comment on this was not addressed.
  3. I suggest the discussion on the implications of their study findings, and the strengths and limitations  should be added to the main discussion section before the "conclusion" heading
  4. I suggest the inclusion of an ethical consideration title just before the "Result" A brief statement about the ethics will be helpful.
  5. Thank you

.

Author Response

Comment 1. The first paragraph of the discussion section looks awkward. There are about four figures and one table within one sentence. I think that is inappropriate in my opinion. Answer. Thank you very much for your professional review of this manuscript. In this revised version, we rewrote the discussion section through the lens of the potential application of this MTM-RTCM mechanism (Section 5.1), and implications and theoretical thinking of the proposed method (Section 5.2). Section 5.1 is focused on how to promote the MTM-RTCM mechanism to five major 2SFCA models. Section 5.2 would illustrate the implications strengths and limitations of our findings, and possible differences in term of policy initiatives if we based our policies on STM instead of MTM-RTMC. What's more, we try to answer what motivates the phenomenon that pediatrics in Nanjing is showing an aggregate developmental effect. Comment 2. I still think the authors should discuss their study findings in comparison with similar studies conducted in other jurisdictions/countries. My previous comment on this was not addressed. Answer. According to your comments, we rewrote the discussion section through the lens of the potential application of this MTM-RTCM mechanism (Section 5.1), and implications and theoretical thinking of the proposed method (Section 5.2). Section 5.1 is focused on how to promote the MTM-RTCM mechanism to five major 2SFCA models. Section 5.2 illustrates the implications strengths and limitations of our findings, and possible differences in term of policy initiatives if we based our policies on STM instead of MTM-RTMC. What's more, we try to answer what motivates the phenomenon that pediatrics in Nanjing is showing an aggregate developmental effect. Comment 3. I suggest the discussion on the implications of their study findings, and the strengths and limitations should be added to the main discussion section before the "conclusion" heading. Answer. Thank you for your reminding. We rewrote the discussion section through the lens of the potential application of this MTM-RTCM mechanism (Section 5.1), and the implications and theoretical thinking of the proposed method (Section 5.2). Section 5.1 is focused on how to promote the MTM-RTCM mechanism to five major 2SFCA models. Section 5.2 would illustrate the implications strengths and limitations of our findings, and possible differences in term of policy initiatives if we based our policies on STM instead of MTM-RTMC. What's more, we try to answer what motivates the phenomenon that pediatrics in Nanjing is showing an aggregate developmental effect. Comment 4. I suggest the inclusion of an ethical consideration title just before the "Result" A brief statement about the ethics will be helpful. Answer. We have added Section 3.3, Ethical consideration, to illustrate all data which are collected from the internet are open source, ethically free, and privacy-free. All of those research data are collected from open-source web approaches and are not involving any personally sensitive information, except the street block shapefile. So the ethical approval is at low risk.

Reviewer 2 Report

I think it would be good to add a discussion comparing  possible  differences in term of policy initiatives if we based our policies on STM instead of MTM-RTM

Author Response

Comment 1. I think it would be good to add a discussion comparing  possible  differences in term of policy initiatives if we based our policies on STM instead of MTM-RTM.

Answer. Thank you for your reminding. We rewrote the discussion section through the lens of the potential application of this MTM-RTCM mechanism (Section 5.1), and implications and theoretical thinking of the proposed method (Section 5.2). Section 5.1 is focused on how to promote the MTM-RTCM mechanism to five major 2SFCA models. Section 5.2 illustrates the implications strengths and limitations of our findings, and possible differences in term of policy initiatives if we based our policies on STM instead of MTM-RTMC. What's more, we try to answer what motivates the phenomenon that pediatrics in Nanjing is showing an aggregate developmental effect.

Reviewer 3 Report

No any further comment

Author Response

Answer. Thank you for your reminding. We have proceeded fine/minor spell check with the manuscript.